# Mollusc Crystallins: Physical and Chemical Properties and Phylogenetic Analysis

Irina N. Dominova * and Valery V. Zhukov

Institute of Medicine and Life Sciences (MEDBIO), Immanuel Kant Baltic Federal University,
236040 Kaliningrad, Russia
* Correspondence: idominova@kantiana.ru; Tel.: +7-921-262-42-19

**Abstract:** The purpose of the present study was to perform bioinformatic analysis of crystallin diversity in aquatic molluscs based on the sequences in the NCBI Protein database. The objectives were as follows: (1) analysis of some physical and chemical properties of mollusc crystallins, (2) comparison of mollusc crystallins with zebrafish and cubomedusa *Tripedalia cystophora* crystallins, and (3) determination of the most probable candidates for the role of gastropod eye crystallins. The calculated average GRAVY values revealed that the majority of the seven crystallin groups, except for μ- and ζ-crystallins, were hydrophilic proteins. The predominant predicted secondary structures of the crystallins in most cases were α-helices and coils. The highest values of refractive index increment (*dn/dc*) were typical for crystallins of aquatic organisms with known lens protein composition (zebrafish, cubomedusa, and octopuses) and for S-crystallin of *Pomacea canaliculata*. The evolutionary relationships between the studied crystallins, obtained from multiple sequence alignments using Clustal Omega and MUSCLE, and the normalized conservation index, calculated by Mirny, showed that the most conservative proteins were Ω-crystallins but the most diverse were S-crystallins. The phylogenetic analysis of crystallin was generally consistent with modern mollusc taxonomy. Thus, α- and S-, and, possibly, J1A-crystallins, can be assumed to be the most likely candidates for the role of gastropod lens crystallins.

**Keywords:** molluscs; crystallins; lens; amino acid composition; conservation; phylogenetic analysis

## 1. Introduction

Crystallins are water-soluble proteins first described as molecular components of the vertebrate eye lens, where they are present in high concentrations [1]. These proteins play a key role in maintaining the transparency of the lens as well as in forming its refractive power. At the same time, crystallins form a diverse group of multifunctional proteins. Based on the size and charge of the protein molecules, as well as their immunological properties, three main groups of vertebrate crystallins are distinguished, α, β, and γ, which presumably originated in their ancestral forms and now constitute the majority of lens proteins found in existing species [2]. In this case, there are specific crystallins that are found only in certain groups of animals. The peculiarity of these crystallins is their structural similarity to and common origin with cytosolic housekeeping enzymes [3]. The S- and Ω-crystallins are mollusc-specific crystallins particularly to cephalopods and bivalves, derived from glutathione S-transferase and aldehyde dehydrogenase, respectively [4]. S-crystallins are the main proteins of the lens in cephalopods. Their amino acid sequences have a high degree (41%) of identity with the sigma-class glutathione-S-transferase (GST) sequence [5]. It is assumed that S-crystallins originated as a result of multiple gene duplications and exon rearrangement [6]. These proteins can be divided into "short-loop" S-crystallins (SL11, S4, and Cry9), which are more similar to GST and have a reduced level of expression in the lens, and "long-loop" S-crystallins, which are predominantly expressed in the lens but are enzymatically inactive [7]. Ω-crystallins are another group of mollusc-specific proteins that

are thought to have originated from a duplication of the ancestral aldehyde dehydrogenase (ALDH) gene [8]. They are cytoplasmic, enzymatically inactive, lens-specific proteins that have the properties of both cytoplasmic and mitochondrial proteins of the ALDH family. It has to be noted that scallop Ω-crystallins are the only crystallins derived from the mitochondrial proteins ALDH2 and ALDH1B1 [9]. Among them, ALDH1A9 is considered the main scallop lens protein (about 70% of all water-soluble proteins). It is a homologue of Ω-crystallins belonging to the class 1/2 ALDH, less numerous crystallins of the cephalopod molluscan lens. The Ω-crystallins of octopus (ALDH1C1) and squid (ALDH1C2) are expressed not only in their eyes, but also in other tissues [10]. In some squids, they are synthesized in photophore lenses, and are called L-crystallins [11]. In addition to the listed proteins, other crystallins and crystallin-like proteins, such as J, λ-, μ-, and ζ, are also found in the genomes of molluscs, and their properties and functions have been described in some animals of other types in which these proteins were first discovered. The functional roles of these proteins in molluscs remain unknown, although the presence of crystallins in the lens of the eye of cephalopods and bivalves suggests their involvement in visual functioning.

There are practically no data on lens crystallins of gastropod molluscs. The only previous work reported the isolation of three polypeptides (80, 63, and 28 kDa), presumably crystallins, from the marine mollusc *Aplysia californica* (J.G. Cooper, 1863) [12,13]. To date, however, this research has not been further developed and has remained incomplete. In the case of terrestrial molluscs, such studies have not been conducted at all. Since crystallins in both vertebrates and invertebrates are expressed not only in the eye, but also in other organs and tissues, the question of what functions these crystallins perform in the molluscan organism remains open. Based on the current knowledge gap regarding the set of crystallins and their functions in molluscs, the main goal of this study was a bioinformatic analysis of the diversity of these proteins in aquatic molluscs based on the amino acid sequences in the NCBI Protein database. The specific objectives were as follows: (1) analysis of some physical and chemical properties of crystallins of several species of aquatic molluscs, (2) comparison of the obtained parameters with known lens crystallins of other aquatic organisms, and (3) determination of the most probable candidates for the role of lens crystallins in gastropods. The zebrafish *Danio rerio* (Hamilton, 1822) and the cubomedusa *Tripedalia cystophora* (Conant, 1897) were chosen as comparison organisms. *D. rerio* was chosen because the data on the qualitative and quantitative composition of the crystallins of this fish lens are the most complete [14], and the cubomedusa was chosen as the organism in which J1 crystallins were first discovered in the lens [15].

## 2. Materials and Methods

### 2.1. Studied Genomes and Crystallin Sequences

For the phylogenetic analysis, representatives of three classes of molluscs with deciphered and annotated genomes were selected, i.e., the cephalopods *Octopus sinensis* (Cuvier, 1797) and *Octopus bimaculoides* (Pickford and McConnaughey, 1949); the bivalves *Pecten maximus* (Linnaeus, 1758) and *Placopecten magellanicus* (Gmelin, 1791); and the gastropods *Aplysia californica* (J.G. Cooper, 1863), *Biomphalaria glabrata* (Say, 1818), and *Pomacea canaliculata* (Lamarck, 1819). The crystallin amino acid sequences were taken from the NCBI Protein database (https://www.ncbi.nlm.nih.gov/protein/ (accessed on 15 July 2022)) and are presented in Table S1.

### 2.2. Analysis of Several Physical and Chemical Properties and Calculation of the Refractive Index Increment of Crystallins

The refractive index increment (*dn/dc*) values of the crystallins of molluscs, *Danio rerio*, and *Tripedalia cystophora* were obtained as weighted average *dn/dc* values predicted from the protein amino acid composition. For this purpose, a model developed by Zhao et al. was used, wherein the refractive index of a protein is fully explained by its amino acid composition [16]. According to this approach, the percentage content of each amino acid was calculated using ProtParam software (https://web.expasy.org/protparam/) on the

ExPASy server (https://www.expasy.org/ (accessed on 15 July 2022)) [17], then multiplied by the individual dn/dc value of each amino acid and summed. Also using ProtParam, the Grand average of hydropathicity index (GRAVY) values were calculated [18]; according to this index, proteins with a GRAVY > 0 are more likely to be hydrophobic proteins and, correspondingly, proteins with a hydrophobicity index below 0 are hydrophilic and most likely cytoplasmic proteins [19].

Prediction of the secondary structure of crystallins was performed using the RaptorX-Property software (http://raptorx.uchicago.edu/) [20], which represents the number of regular (α-helixes, β-sheets) and irregular secondary structures (coils) as a percentage.

Graphs of amino acid composition, GRAVY indices, secondary structures, and *dn/dc* crystallins were plotted using GraphPad Prism 9 (GraphPad Software, Inc., San Diego, CA, USA).

### 2.3. Multiple Sequence Alignment and Phylogenetic Analysis

Crystallin families expressed in all three classes of molluscs were selected for amino acid sequence alignment.

Alignment was performed using the Unipro UGENE software (version 39.0) (Unipro, Russia) [21]. Clustal Omega and MUSCLE algorithms integrated in UGENE were used to align the amino acid sequences. Initial alignment was performed by Clustal Omega with the following parameters: number of iterations—100, max number guide-tree iterations—100, max number of HMM (Hidden Markov Models) iterations—100; it was then refined by the MUSCLE using whole amino acid sequences, run with a max number of iterations equal to 100. The Clustal Omega and MUSCLE algorithms were chosen for alignment because they are among the fastest and most accurate algorithms, allowing efficient multiple alignments of both amino acid and nucleotide sequences [22,23]. After the initial alignment to construct the final multiple alignments and phylogenetic trees, some sequences with low quality were removed, in particular XP_013065813.1 Crystallin J1A-like (*B. glabrata*), XP_013065454.1 Crystallin J1A-like (*B. glabrata*), XP_036356168.1 Lambda-crystallin homolog isoform X3 (*O. sinensis*), XP_013066599.1 PREDICTED: S-crystallin SL11-like (*B. glabrata*), XP_014771126.1 S-crystallin 3 isoform X1 (*O. bimaculoides*), XP_014781642.1 S-crystallin 4-like isoform X2 (*O. bimaculoides*), and XP_014772690.1 S-crystallin 4-like (*O. bimaculoides*).

The alignment results, namely, the most conservative regions of the sequences, were visualized using the Jalview software (https://www.jalview.org/ (accessed on 15 July 2022)) [24]. Also using this software, we calculated a normalized conservation index for full sequences according to Mirny [25], which allowed us to measure the entropy index, taking into account the stereochemical properties of amino acids [26], and classify the sequences into seven basic types based on the Taylor diagram [27], using the AACon resource (http://www.compbio.dundee.ac.uk/aacon/index.html (accessed on 15 July 2022)) [28]. The sequences with an average conservation index of more than 0.8 were considered to be highly conservative, those with an index of 0.6–0.8 were considered semiconservative sequences, and lowly conservative sequences were defined as those with an index of less than 0.6.

Phylogenetic analysis was performed using the MEGA X software (version 10.2.6) (https://www.megasoftware.net/ (accessed on 1 July 2022)) [29] via Maximum Likelihood and the Le and Gascuel (LG) corrections model [30] with 1000 bootstrap replications [31]. The evolutionary model was selected using the IQ-TREE Web Server (http://iqtree.cibiv.univie.ac.at/ (accessed on 15 August 2022)) [32], which automatically determined the best-fit model for each alignment. The initial tree for the heuristic search was obtained automatically by Maximum Parsimony. A discrete gamma distribution (+G) with five categories was used to model different evolutionary rates between sites. A rate variation model (+I) was added to the analysis, which allowed some sites to be evolutionarily unchanged. The resulting phylogenetic tree was automatically selected based on the topology with the highest logarithmic likelihood value. The phylogenetic trees were visu-

alized using FigTree software (version 1.4.4) (http://tree.bio.ed.ac.uk/software/figtree/ (accessed on 15 July 2022)) [33].

Gastropod mollusc crystallins were BLAST searched (blastp algorithm) against cephalopod and bivalve crystallins deposited in NCBI GenBank's protein database (https://blast.ncbi.nlm.nih.gov/Blast.cgi (accessed on 12 September 2022)).

## 3. Results

Analysis of the NCBI Gene (https://www.ncbi.nlm.nih.gov/gene/ (accessed on 15 July 2022)) and Protein (https://www.ncbi.nlm.nih.gov/protein/ (accessed on 15 July 2022)) databases showed that in the genomes of *Octopus sinensis*, *Octopus bimaculoides*, *Pecten maximus*, *Placopecten magellanicus*, *Aplysia californica*, *Biomphalaria glabrata*, and *Pomacea canaliculata*, several classes of crystallins and crystallin-like (homologous) proteins quantitatively dominated, in particular α-, J1A-, λ-, μ-, and S-crystallins. However, there were groups of crystallins that were fully or partially specific for certain classes of molluscs. For example, the scallop-specific Ω-crystallins were absent from the genomes of gastropod molluscs but were found in cephalopods, and ζ-like crystallins were found only in cephalopod molluscs and one gastropod species (*B. glabrata*). A single β-crystallin-like protein was found in bivalves (Table S1).

A total of 124 amino acid sequences were selected for analysis, from which the β-crystallin-like protein was excluded because it had no homologues.

### 3.1. Analysis of the Amino Acid Composition and Calculation of the Refractive Index Increment of Crystallins

The refractive index increment *dn/dc* values for aquatic organism lens crystallins are markedly higher than for proteins with similar secondary structures and domains localized in other tissues and organs [34]. This effect is explained by the high proportion of amino acids with high polarizability in lens crystallins [35]. The amino acids with the highest degree of polarizability are arranged in the following order: Trp > Phe > Tyr > His > Arg = Cys > Met [16]. We analyzed the contents of amino acids of this series in mollusc crystallins, as well as in four jellyfish crystallins and nine lens crystallins of adult *D. rerio* (Table S1), because their presence in lens crystallins was confirmed [36,37].

Figure 1 shows the average percentage of amino acids with high polarizability in crystallins. In α-crystallins (Figure 1a,h), highly polarized amino acids accounted for approximately 25% of all amino acids. At the same time, α-crystallins of both molluscs (Figure 1a) and *D. rerio* (Figure 1h) were characterized by quite a high content of Arg (about 8%), except for α-crystallins of *B. glabrata*, in which Arg on average accounted for slightly more than 4% of all amino acids. Additionally, α-crystallins contained a relatively large amount of Phe, for example, 6.5% in *A. californica*, and more than 7% in *D. rerio*. Meanwhile, low levels of Tyr (about 1.5%), Trp (about 1%), and Cys (about 0.6%) prevailed in this group of crystallins, especially in molluscs.

A different situation was observed in the amino acid composition of J1A-crystallins of molluscs (Figure 1b). The proportion of all analyzed amino acids in them was 18%, which was markedly less compared not only with α-crystallins of molluscs, but also with J-crystallins of *D. rerio* (33.8%) and *T. cystophora* (24.9%). The proportion of Arg (4.5%) and Cys (1.8%) was increased and that of Phe (3%) was decreased. The highest proportion of Cys was found in J1A-crystals of *P. maximus* at 2.5%. At the same time, the contents of all highly polarized amino acids in J-crystals of *D. rerio* (Figure 1h) were 2 times higher than in J1A-crystals of molluscs, except for His (1.5 times more) and Met (almost the same).

The proportion of polarized amino acids in λ-crystallins (Figure 1c) was similar to that in J1A-crystallins, except for a lower His content of 1.4% and a slightly higher Arg proportion (4.9%). In μ-crystallins (Figure 1d), the proportion of highly polarized amino acids was 15%. The Phe content (3.4%) in them approximately corresponded to that in J1A- and λ-crystallins, whereas the Arg content was reduced to 2.3% and Cys content to 1.5%. The Ω-crystallins (Figure 1e) consisted of about 20% of the most polarizable amino acids

and their composition was generally similar to that of the crystallins already considered. However, they differed in the His content (2.2% in cephalopods versus 0.6% in bivalves) and Tyr content (4.7% in cephalopods versus 2.8% in bivalves).

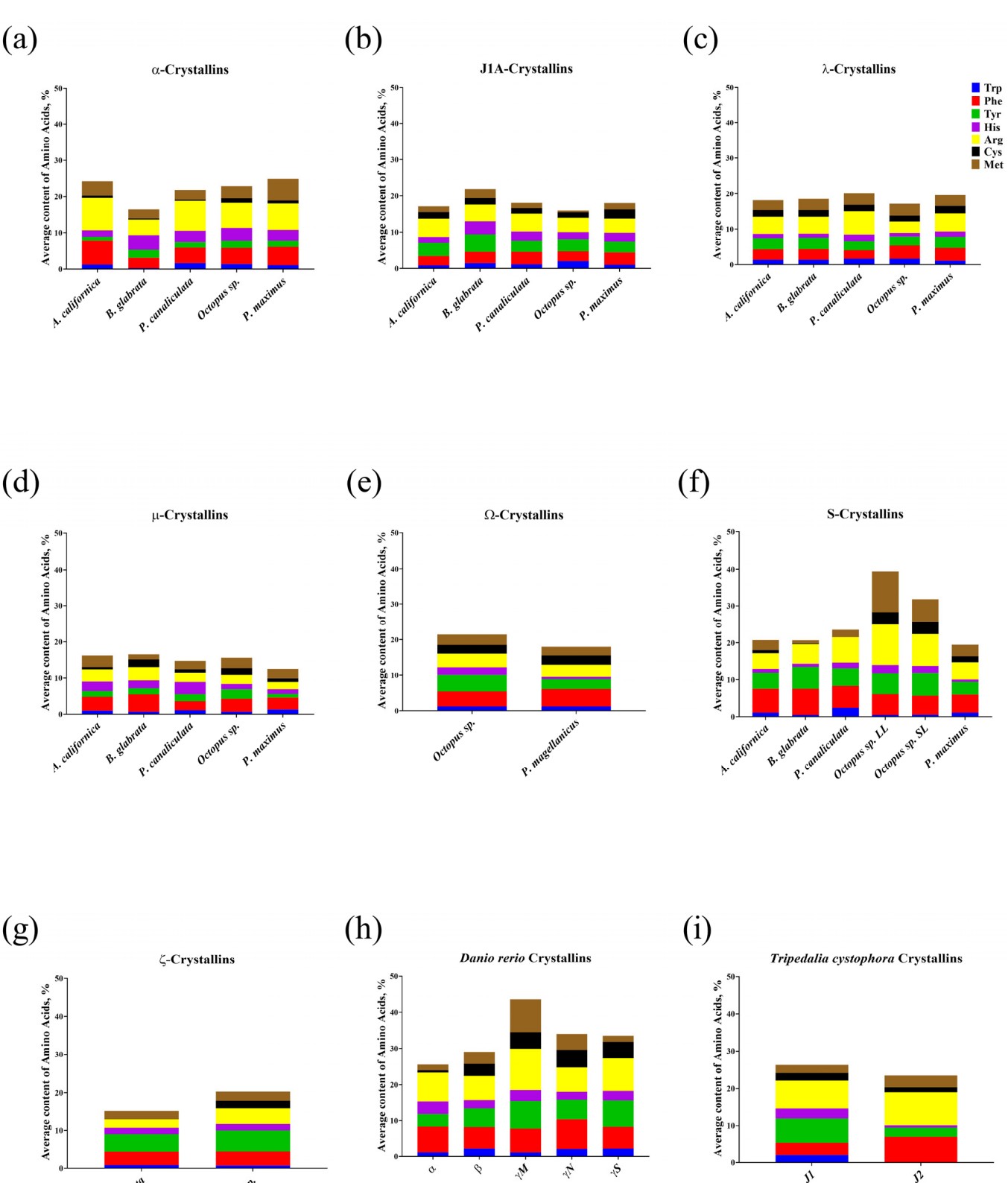

**Figure 1.** Average percentage contents of the most highly polarized amino acids in the sequences of mollusc crystallins and lens crystallins of *Danio rerio* and *Tripedalia cystophora*.

Analysis of the amino acid composition of S-crystallins (Figure 1f) revealed high contents of Met (8.8%) and Arg (9.9%) in the cephalopod molluscs, for which they were the main lens proteins. In addition, the proportion of highly polarizable amino acids in these molluscs was almost 36%, which was comparable with the lens crystallins of *D. rerio* (Figure 1h). S-crystallins of gastropods and bivalves contained significantly less highly polarizable amino acids (about 21%) and the percentages of these amino acids were quite close to each other. In addition, a common feature of all S-crystallins was the low content of Trp (except for *P. caniculata*), His (especially in *P. maximus*—0.6%), and Cys, which was completely absent in *P. caniculata* and had a minimal value in *B. glabrata*—0.2%.

The ξ-crystallins (Figure 1g), found only in cephalopod molluscs and *B. glabrata*, were characterized by an average content of highly polarizable amino acids (15% and 20%, respectively) relative to other families of crystallins. At the same time, the ξ-crystallins of the gastropod mollusc lacked Cys. The ratios of other amino acids varied insignificantly.

Analysis of the calculated values of the increment of refractive index of mollusc crystallins and lens crystallins of *D. rerio* and *T. cystophora* is shown in Figure 2. The γM-crystallins (*dn/dc* = 0.197 mL/g) of *D. rerio* (Figure 2h), which, as known, occupy the dominant position among fish lens proteins [38], had the highest increment of refractive index, and it is supposed these proteins determine the special optical properties of these aquatic organism lens. The closest in value of the increment of refractive index to the γM-crystallins of *D. rerio* were the S-crystallins (*dn/dc* = 0.193 mL/g) of cephalopod molluscs (Figure 2f), which, as already mentioned above, form the basis of their lens [5]. J1-crystallins (*dn/dc* = 0.190 mL/g) of the cubomedusa *T. cystophora* also had similar values of the increment of refractive index (Figure 2i). For other proteins, Ω-crystallins (Figure 2e), reliably expressed in the lens of bivalves [10], showed an increment of refractive index value that was significantly lower (*dn/dc* = 0.186 mL/g) not only compared to *D. rerio* crystallins, but also compared to the Ω-crystallins of octopus (*dn/dc* = 0.188 mL/g). Additionally, S-crystallins of molluscs in principle had the highest values of the increment of refractive index (averaged *dn/dc* = 0.189 mL/g), whereas the lowest values of the increment of refractive index of molluscs were in μ-crystallins (averaged *dn/dc* = 0.183 mL/g) (Figure 2d). The remaining mollusc crystallins had quite similar increment of refractive index values to each other.

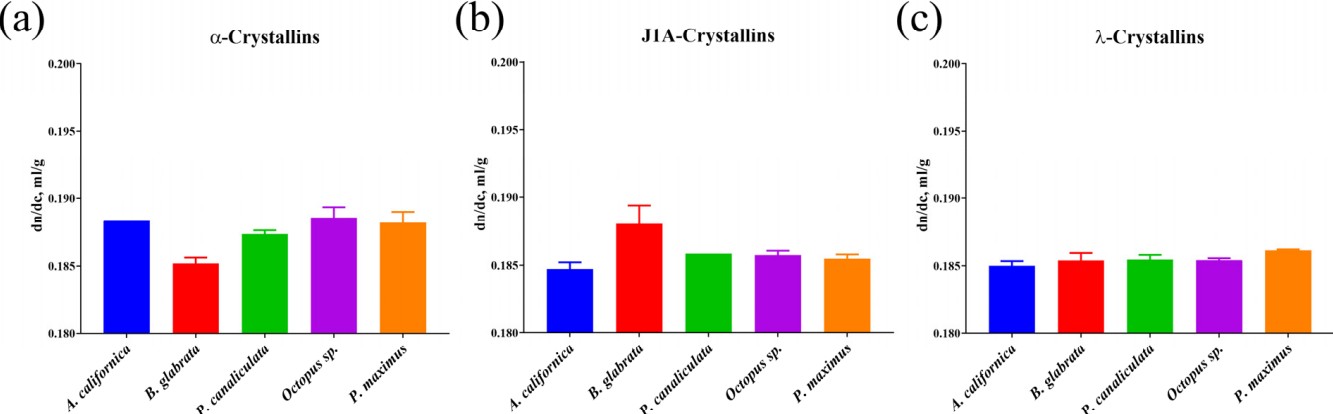

**Figure 2.** *Cont.*

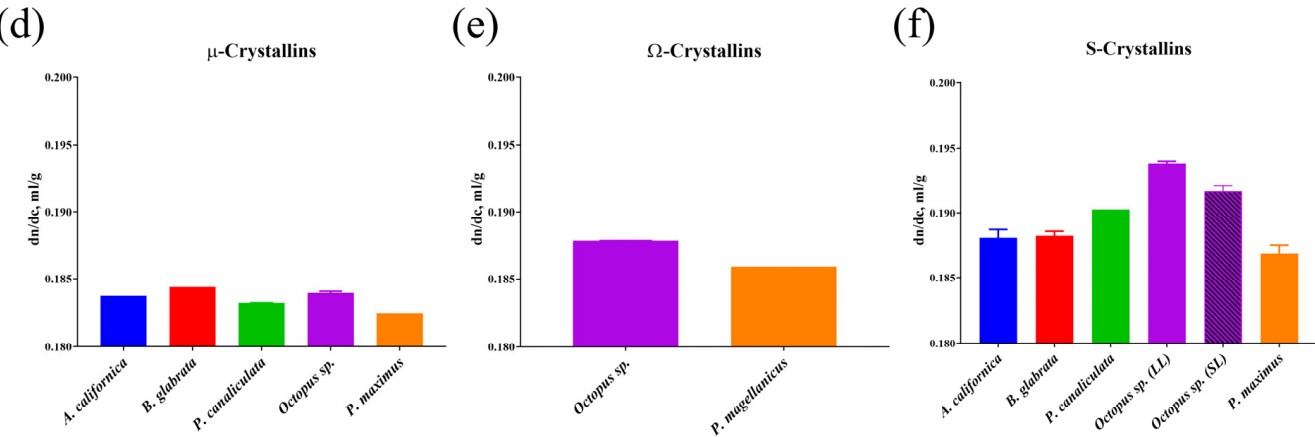

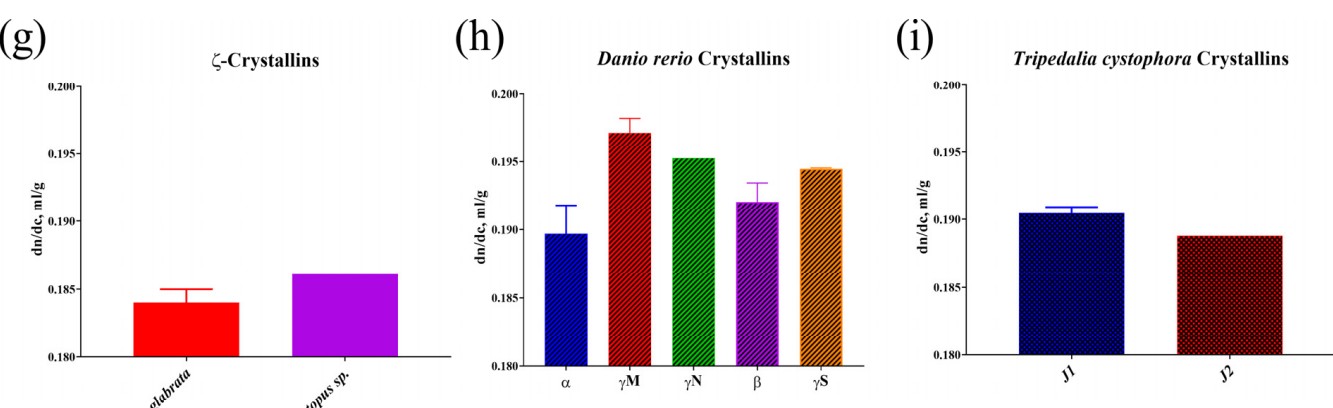

**Figure 2.** Calculated mean values of the increment of refractive index (*dn/dc*, mL/g) for the groups of mollusc crystallins and lens crystallins of *Danio rerio* and *Tripedalia cystophora*. Data are presented as mean ± SEM (except for n = 1).

### 3.2. Crystalline Hydrophobicity Analysis

Most mollusc crystallins, as well as lens crystallins of *D. rerio* and *T. cystophora* (Figure 3), had values of the Grand average of hydropathicity index (GRAVY) below zero and, therefore, were determined to be hydrophilic cytoplasmic proteins, except for μ- and ζ-crystallins (Figure 3d,g), which had GRAVY index values significantly above 0, indicating their hydrophobic character and, most likely, intramembrane topography. The GRAVY value for the single μ-crystallin of *A. californica*, although below zero, was still not sufficient to consider it a hydrophilic protein. Crystallins of *D. rerio* (Figure 3h), *T. cystophora* (Figure 3i), and mollusc α-crystallins (Figure 3a) had the lowest values of the GRAVY index and hence the highest hydrophilicity, followed by S-crystallins of octopus and *P. canaliculata* (Figure 3f). All other mollusc crystallins had hydrophobicity index (GRAVY) values that were reasonably close to each other.

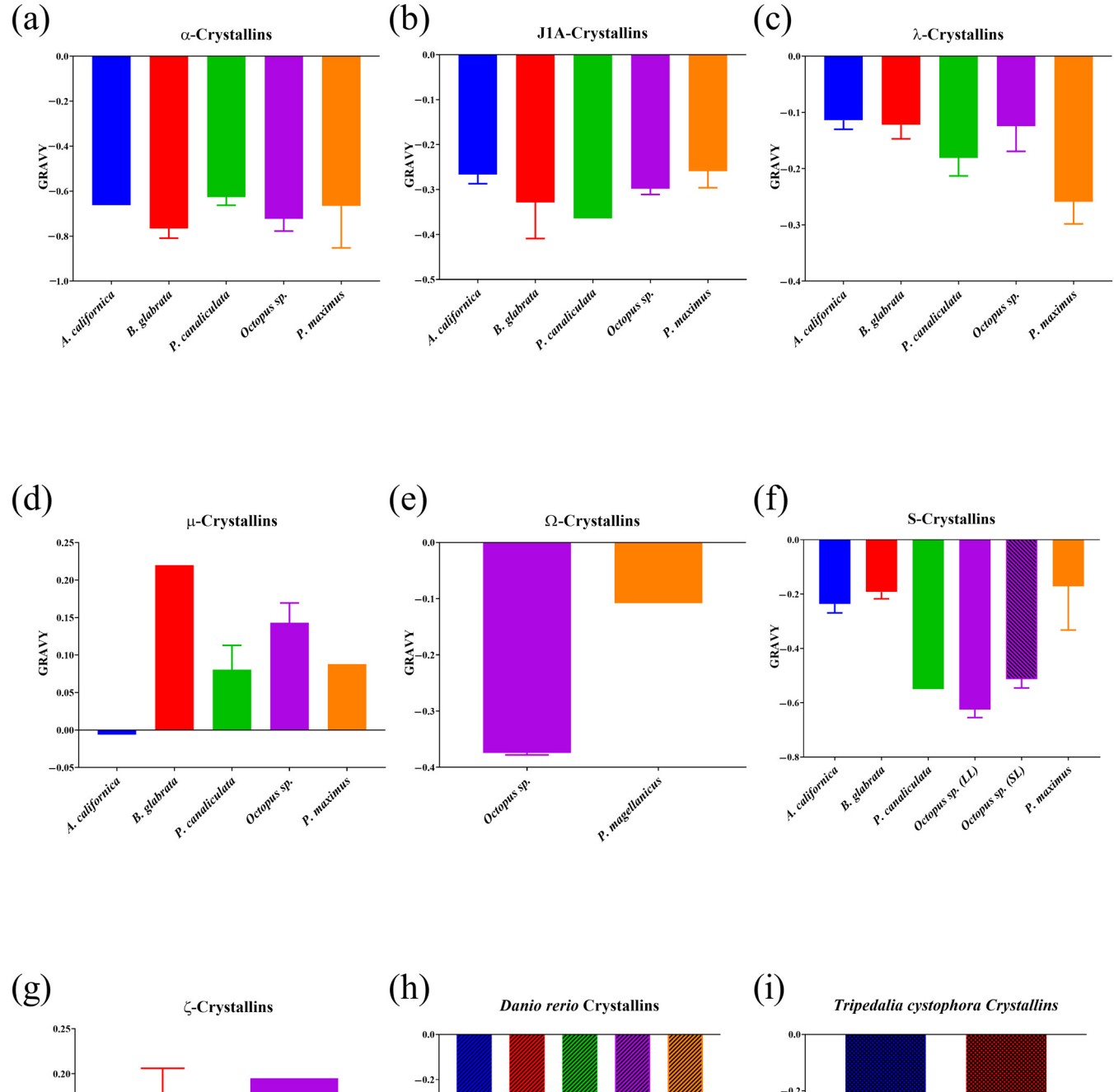

**Figure 3.** Mean values of the Grand average of hydropathicity index (GRAVY) for mollusc crystallins and lens crystallins of *Danio rerio* and *Tripedalia cystophora*. Data are presented as mean ± SEM (except for n = 1).

### 3.3. The Secondary Structure Analysis of Crystallins

All analyzed crystallins (Figure 4) were characterized by the arrangement of sequences in α-helices and coils, except for mollusc α-crystallins (Figure 4a) and *D. rerio* lens crystallins (Figure 4h), which had almost absent α-helices in the secondary structure. In the secondary structures of J1-crystallins of both molluscs (Figure 4b) and *T. cystophora* (Figure 4i), β-sheets were almost completely absent. The percentage of β-sheets in the S-crystallins of molluscs was also quite low (Figure 4f). All other mollusc crystallins have quite similar predicted secondary structures.

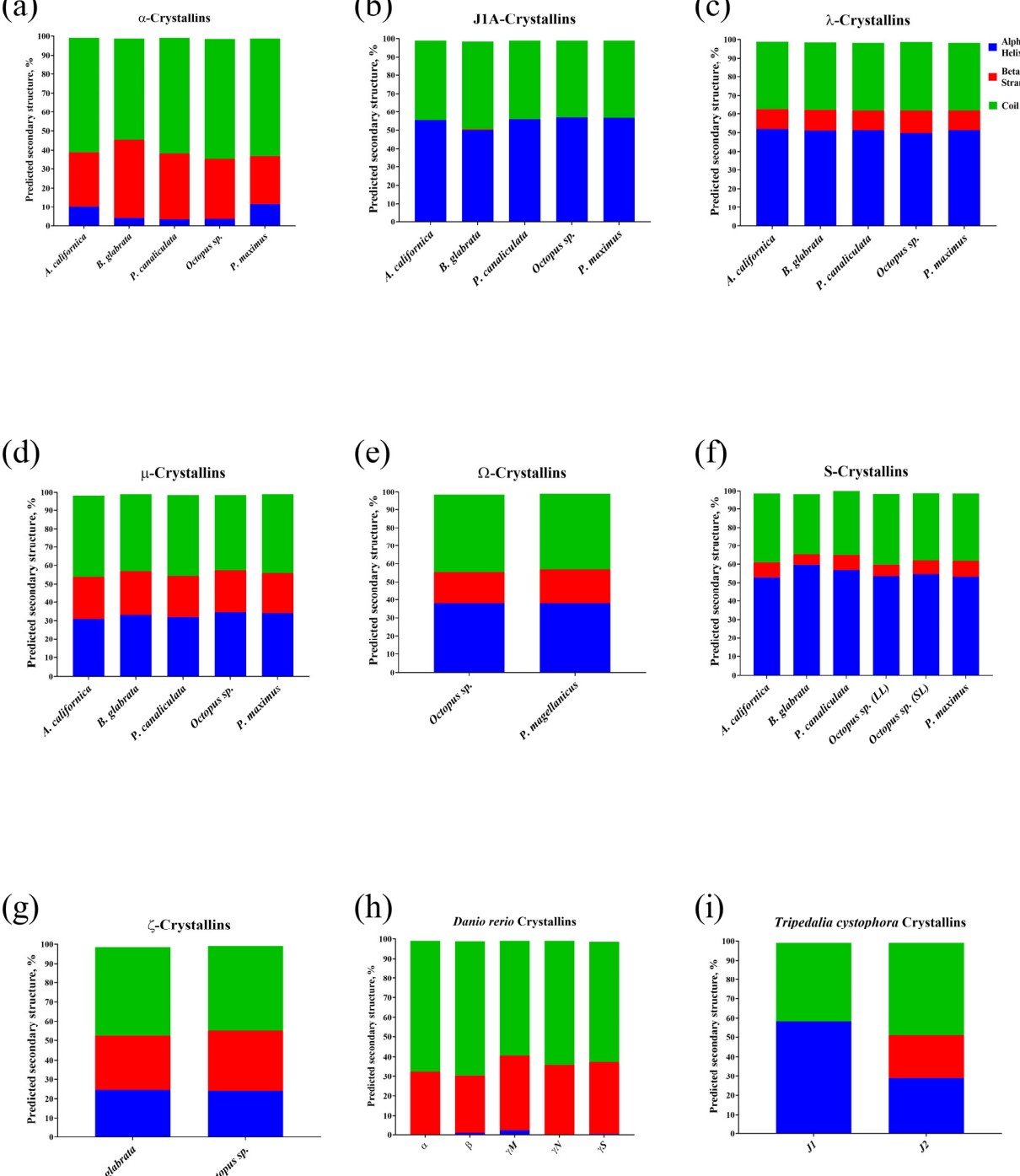

**Figure 4.** Mean values of the predicted secondary structures in percentages for the mollusc crystallins and lens crystallins of *Danio rerio* and *Tripedalia cystophora*.

### 3.4. Multiple Crystallin Sequence Alignment Analysis

Analysis of the amino acid sequences of crystallins and crystallin-like proteins encoded in mollusc genomes showed that their mean normalized conservation index by Mirny was 0.73 (Tables 1 and S2). At the same time, their structures contained highly conserved motifs which were typical for each specific group of crystallins. The most conservative proteins were Ω-crystallins; their sequences contained 55% identical amino acids and 23.5% conservative substitutions. ζ-crystallins were a less conservative group of proteins according to Mirny's algorithm and S-crystallins showed the greatest sequence diversity: only one identical amino acid, four conservative substitutions, and six semiconservative ones.

**Table 1.** Analysis of multiple sequence alignment of crystallins and crystallin-like proteins of molluscs.

| Crystallins | Number of Analyzed Sequences | The Length of the Analyzed Sequences, in Amino Acid Residues | Normalized Conservation Index by Mirny, Mean $\pm$ SD | Number of Identical Amino Acids | Number of Conservative Substitutions | Number of Semi-Conservative Substitutions |
|---|---|---|---|---|---|---|
| α-crystallins | 23 | Minimum: 87 Maximum: 233 Average: 156 | 0.83 $\pm$ 0.26 | 2 | 19 | 7 |
| J1A-crystallins | 9 | Minimum: 254 Maximum: 334 Average: 286 | 0.68 $\pm$ 0.26 | 51 | 32 | 20 |
| λ-crystallins | 19 | Minimum: 298 Maximum: 333 Average: 320 | 0.65 $\pm$ 0.29 | 30 | 52 | 20 |
| μ-crystallins | 9 | Minimum: 250 Maximum: 312 Average: 287 | 0.72 $\pm$ 0.27 | 72 | 46 | 27 |
| Ω-crystallins | 3 | Minimum: 492 Maximum: 495 Average: 494 | 0.85 $\pm$ 0.25 | 274 | 117 | 33 |
| S-crystallins | 51 | Minimum: 182 Maximum: 318 Average: 220 | 0.75 $\pm$ 0.25 | 1 | 4 | 6 |
| ζ-crystallins | 3 | Minimum: 105 Maximum: 323 Average: 239 | 0.64 $\pm$ 0.20 | 34 | 25 | 9 |

The comparative analysis of the amino acid sequences revealed no strict regularity in the localization of the conserved regions and the degrees of their hydrophobicity/hydrophilicity. Thus, the most conserved regions in α-crystallins were closer to the C-end. Hydrophilic amino acids quantitatively prevailed over hydrophobic ones (26 vs. 12) (Figure S1a). In contrast, in J1A-crystallin molecules, conserved regions were located in the first two thirds of the sequences (Figure S1b), but also contained more hydrophilic amino acid residues (64 versus 35). A very different situation was observed in λ-crystallin molecules, in which conservative regions were evenly distributed along the entire length of the molecule except for the C-terminal itself and were enriched with hydrophilic amino acid residues (Figure S1c). In μ-crystallin molecules, the conserved regions were located in the last two thirds of the sequences and contained approximately equal amounts of hydrophobic and hydrophilic amino acid residues (Figure S1d). Figure S1e shows a fragment of the multiple sequence alignment of Ω-crystallins that contained more hydrophilic amino acid residues than hydrophobic ones in their structure. Conservative regions of S-crystallins, like those of λ-crystallins, were evenly distributed along the entire length of the whole molecule; however, in contrast to λ-crystallins, they were not distinctly hydrophobic

(Figure S1f). In ζ-crystallins, the conserved region was located in the center of the sequence and also contained approximately equal numbers of hydrophilic and hydrophobic amino acids (Figure S1g).

### 3.5. Phylogenetic Analysis of Crystallins

Phylogenetic trees were constructed for α-, J1A-, λ-, μ-, and S-crystallins and crystallin-like proteins; Ω- and ζ-crystallins were excluded because these groups consisted of only three representatives.

The phylogenetic tree (Figure 5) of α-crystallins and crystallin-like proteins shows that proteins of cephalopods and some gastropods (*B. glabrata* and *P. canaliculata*) are arranged in five major clades. The proteins of bivalves and *A. californica* come out of common nodes, and there is also a common node between one protein of *P. canaliculata* and *P. maximus*. Another interesting fact is that the gastropods' α-crystallins are significantly distant from each other. Thus, the proteins of *P. canaliculata* are phylogenetically closer to those of cephalopods than to those of *B. glabrata* and, especially, *A. californica*.

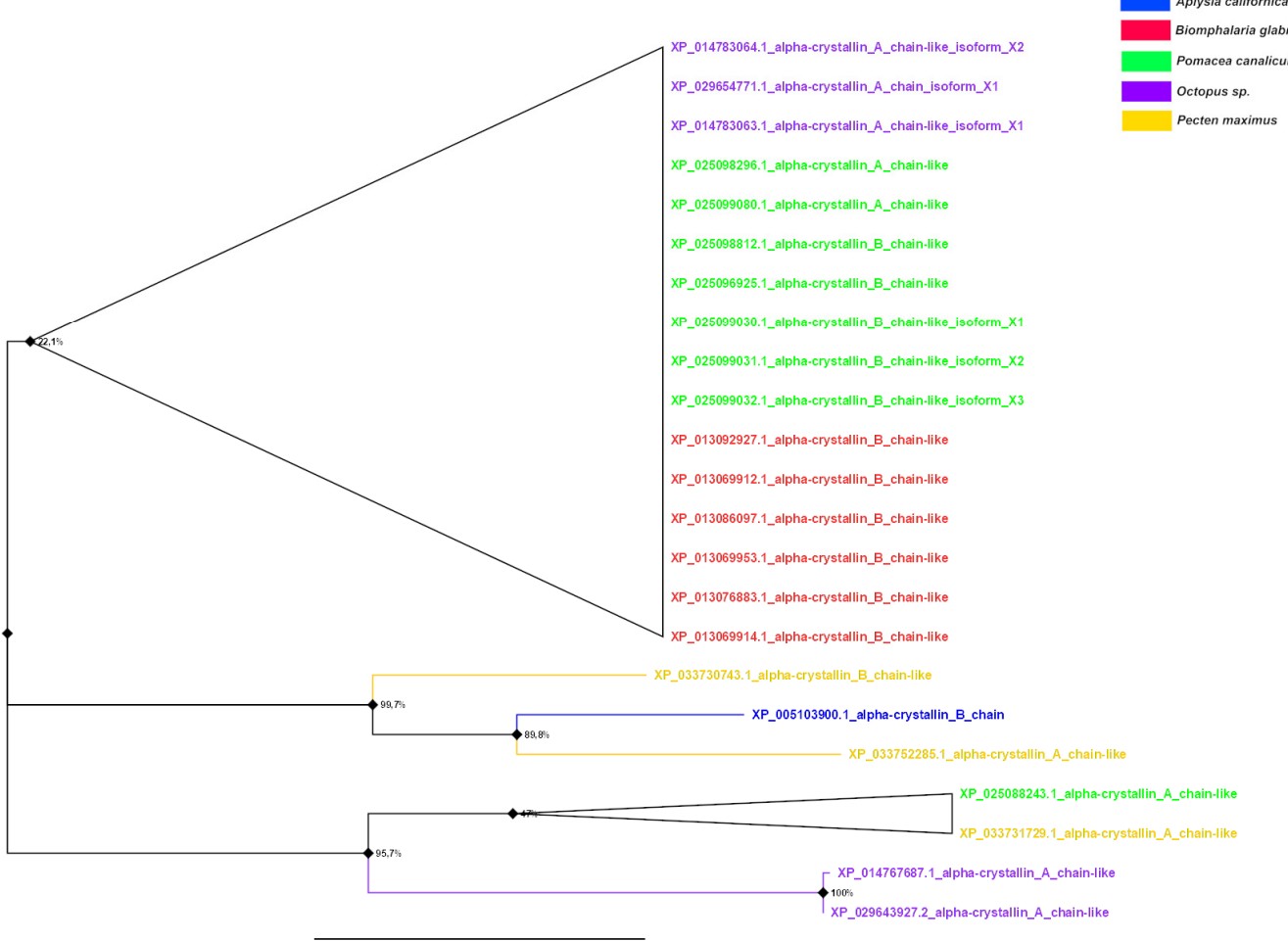

**Figure 5.** Phylogenetic tree of α-crystallins of molluscs.

A tree was constructed for 23 amino acid sequences (the final dataset had a total of 97 positions) using the Maximum Likelihood (ML) statistical method and the LG correction model with 1000 bootstrap repeats. A discrete Gamma distribution was used to model evolutionary rate differences among sites (four categories (+G, parameter = 2.4206)). The rate variation model allowed for some sites to be evolutionarily invariable ([+I], 1.03% sites). The tree is drawn to scale, with branch lengths measured in the number of substitutions

per site. Bootstrap values in % are given in nodes. The nodes with bootstrap support <50% have been collapsed.

J1-crystallins and crystallin-like proteins showed less diversity compared to α-crystallins not only in the number of representatives of this group, but also in the length of the phylogenetic tree branches (Figure 6). The proteins of cephalopods and bivalves formed two clades on the tree. Moreover, one of the proteins of the latter clustered with the J1A-crystallins of gastropod molluscs. At the same time, bivalve mollusc proteins were the most distant proteins from all the others. J1A-crystallins of gastropods were quite similar, except for the J1A-crystallin-like protein of *P. canaliculata*, which also, as in the case of α-crystallins, did not unite in common nodes with proteins of *A. californica*.

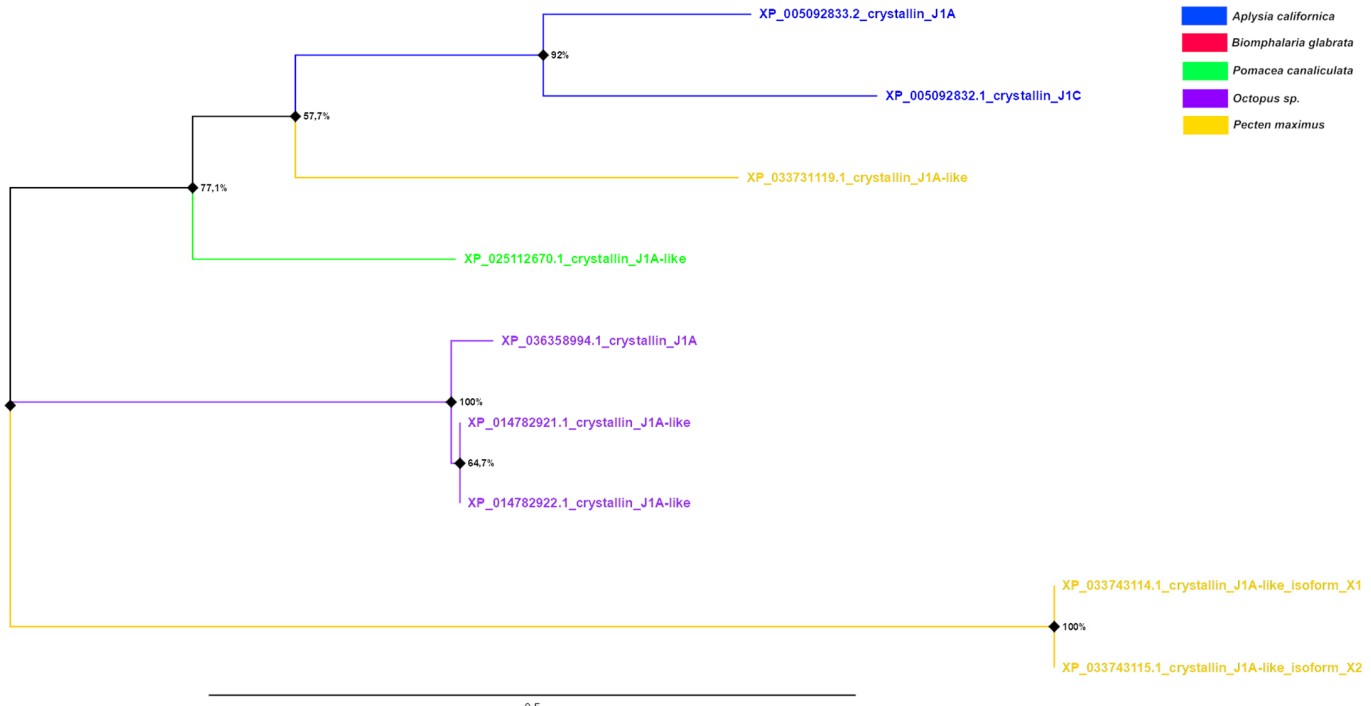

**Figure 6.** Phylogenetic tree of J1-crystallins of molluscs.

A tree was constructed for nine amino acid sequences (the final dataset had a total of 189 positions) using the Maximum Likelihood (ML) statistical method and the LG correction model with 1000 bootstrap repeats. A discrete Gamma distribution was used to model evolutionary rate differences among sites (four categories (+G, parameter = 2.5490)). The rate variation model allowed for some sites to be evolutionarily invariable ([+I], 12.25% sites). The tree is drawn to scale, with branch lengths measured in the number of substitutions per site (above the branches). Bootstrap values in % are given in nodes. The nodes with bootstrap support <50% have been collapsed.

Analysis of the λ-crystallin phylogenetic tree (Figure 7) showed that the proteins of cephalopod molluscs formed a single clade, whereas the crystallins of other molluscs had common nodes with each other. In addition to the λ-crystallins of cephalopod molluscs, the proteins of *A. californica*, *P. canaliculata*, and bivalve molluscs formed clades. Among the gastropod molluscs, the four crystallins of *A. californica* were the most similar to each other. Furthermore, the λ-crystallins of *A. californica* shared common nodes with the proteins of *B. glabrata*, whereas the two crystallins of *P. canaliculata* formed a separate group quite distant from other gastropod molluscs. The proteins of bivalves were phylogenetically predominantly closer to the gastropods than to the cephalopods.

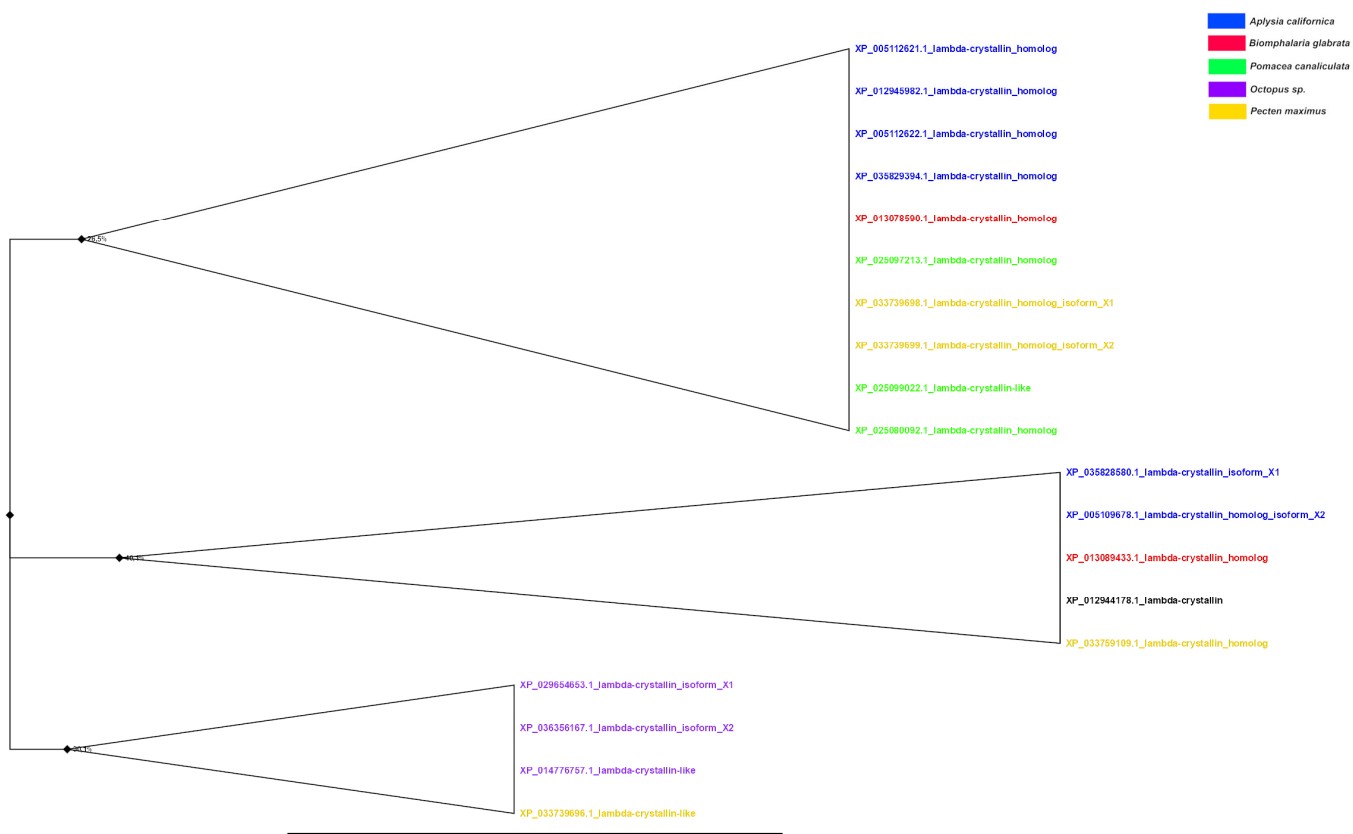

**Figure 7.** Phylogenetic tree of λ-crystallins of molluscs.

A tree was constructed for 19 amino acid sequences (the final dataset had a total of 312 positions) using the Maximum Likelihood (ML) statistical method and the LG correction model with 1000 bootstrap repeats. A discrete Gamma distribution was used to model evolutionary rate differences among sites (four categories (+G, parameter = 3.0677)). The rate variation model allowed for some sites to be evolutionarily invariable ([+I], 5.45% sites). The tree is drawn to scale, with branch lengths measured in the number of substitutions per site (above the branches). Bootstrap values in % are given in nodes. The nodes with bootstrap support <50 have been collapsed.

On the phylogenetic tree of μ-crystallins (Figure 8), in contrast, a clear distinction into separate classes of proteins based on their belonging to mollusc classes can be seen. The largest clade is formed by the μ-crystallins of cephalopod molluscs. Additionally, the proteins of gastropod molluscs have a common node, although they are distant from each other by a considerable distance.

A tree was constructed for nine amino acid sequences (the final dataset had a total of 251 positions) using the Maximum Likelihood (ML) statistical method and the LG correction model with 1000 bootstrap repeats. A discrete Gamma distribution was used to model evolutionary rate differences among sites (four categories (+G, parameter = 1.9765)). The rate variation model allowed for some sites to be evolutionarily invariable ([+I], 15.14% sites). The tree is drawn to scale, with branch lengths measured in the number of substitutions per site (above the branches). Bootstrap values in % are given in nodes. The nodes with bootstrap support <50 have been collapsed.

Among the analyzed proteins, S-crystallins (Figure 9) represented the most extensive group. In cephalopods, they formed a separate branch in the phylogenetic tree, except for two proteins (XP 014771120.1 S-crystallin 4-like and XP 036358740.1 S-crystallin 4-like) which did not belong to this branch. In cephalopods, S-crystallins belonging to different protein groups (with short and long loops) predominantly formed separate subclades. For example, SL11- and SL20-crystallins were on different branches from other S-crystallins. At

the same time, it was impossible to identify distinct subclades of S1, S2, S3, and S4 crystallins. It was also impossible to identify distinct clades for *A. californica*, *B. glabrata*, and *P. canaliculata* proteins, the S-crystallins of which shared common nodes on the tree. At the same time, the bivalve proteins were organized into a single clade. Thus, the entire phylogenetic tree of S-crystallins can be conditionally divided into two parts, of which one contains proteins of cephalopod molluscs and the other contains proteins of all other organisms.

A tree was constructed for 51 amino acid sequences (the final dataset had a total of 194 positions) using the Maximum Likelihood (ML) statistical method and the LG correction model with 1000 bootstrap repeats. A discrete Gamma distribution was used to model evolutionary rate differences among sites (four categories (+G, parameter = 2.1370)). The rate variation model allowed for some sites to be evolutionarily invariable ([+I], 0.52% sites). The tree is drawn to scale, with branch lengths measured in the number of substitutions per site (above the branches). Bootstrap values in % are given in nodes. The nodes with bootstrap support <50 have been collapsed.

In addition to phylogenetic analysis, we performed separate BLAST searches for *Gastropoda* crystallins against *Cephalopoda* and *Bivalvia* mollusc crystallins to determine the percentage of identity of these amino acid sequences (Table 2).

**Table 2.** Percent identity of *Gastropoda* mollusc crystallins versus *Cephalopoda* and *Bivalvia* mollusc crystallins by BLAST.

| Gastropoda molluscs | Crystallin | Mollusc Protein, GenBank Accession Number | Percent Identity, % | E-Value |
|---|---|---|---|---|
| | | versus *Cephalopoda* | | |
| *Aplysia californica* | α (n = 1) | XP_029637045.1 *O. sinensis* | 56.02 | $4 \times 10^{-56}$ |
| | | XP_014782498.1 *O. bimaculoides* | 55.15 | $3 \times 10^{-56}$ |
| | | XP_029642860.1 *O. sinensis* | 43.37 | $2 \times 10^{-21}$ |
| | | XP_014783828.1 *O. bimaculoides* | 43.37 | $2 \times 10^{-21}$ |
| | | XP_014783827.1 *O. bimaculoides* | 43.37 | $3 \times 10^{-21}$ |
| | J1A (n = 2) | XP_014782921.1 *O. bimaculoides* | 44.81 | $4 \times 10^{-72}$ |
| | | XP_036358994.1 *O. sinensis* | 42.98 | $7 \times 10^{-70}$ |
| | λ (n = 7) | XP_029654653.1 *O. sinensis* | 45.64 | $3 \times 10^{-102}$ |
| | | XP_014776757.1 *O. bimaculoides* | 45.30 | $7 \times 10^{-102}$ |
| | | XP_036356167.1 *O. sinensis* | 45.14 | $6 \times 10^{-97}$ |
| | | XP_036356168.1 *O. sinensis* | 44.91 | $1 \times 10^{-69}$ |
| | | XP_014787243.1 *O. bimaculoides* | 28.81 | $5 \times 10^{-17}$ |
| | | XP_014772028.1 *O. bimaculoides* | 47.93 | $3 \times 10^{-76}$ |
| | μ (n = 1) | XP_036359829.1 *O. sinensis* | 47.52 | $1 \times 10^{-74}$ |
| | | XP_014772027.1 *O. bimaculoides* | 45.61 | $2 \times 10^{-90}$ |
| | | XP_029637902.1 *O. sinensis* | 45.27 | $3 \times 10^{-88}$ |
| | S short loop (n = 6) | XP_014771120.1 *O. bimaculoides* | 38.89 | $7 \times 10^{-18}$ |
| | | XP_014771119.1 *O. bimaculoides* | 38.38 | $2 \times 10^{-15}$ |
| | | XP_036355061.1 *O. sinensis* | 35.54 | $6 \times 10^{-18}$ |
| | | XP_014772688.1 *O. bimaculoides* | 33.80 | $5 \times 10^{-16}$ |
| | | XP_036359314.1 *O. sinensis* | 33.17 | $2 \times 10^{-22}$ |

| Gastropoda molluscs | Crystallin | Mollusc Protein, GenBank Accession Number | Percent Identity, % | E-Value |
|---|---|---|---|---|
| *Biomphalaria glabrata* | α (n = 6) | XP_014783064.1 *O. bimaculoides* | 52.63 | $1 \times 10^{-28}$ |
| | | XP_014783063.1 *O. bimaculoides* | 52.63 | $1 \times 10^{-28}$ |
| | | XP_029654771.1 *O. sinensis* | 51.58 | $4 \times 10^{-28}$ |
| | | XP_036354796.1 *O. sinensis* | 42.75 | $4 \times 10^{-28}$ |
| | | XP_029637045.1 *O. sinensis* | 41.94 | $3 \times 10^{-15}$ |
| | J1A (n = 2) | XP_014782921.1 *O. bimaculoides* | 43.06 | $9 \times 10^{-40}$ |
| | | XP_036358994.1 *O. sinensis* | 41.67 | $7 \times 10^{-38}$ |
| | λ (n = 2) | XP_036356167.1 *O. sinensis* | 48.34 | $3^{-110}$ |
| | | XP_029654653.1 *O. sinensis* | 47.92 | $1 \times 10^{-114}$ |
| | | XP_014776757.1 *O. bimaculoides* | 47.60 | $2 \times 10^{-115}$ |
| | | XP_036356168.1 *O. sinensis* | 44.91 | $3 \times 10^{-70}$ |
| | | XP_014769218.1 *O. bimaculoides* | 29.17 | $4 \times 10^{-12}$ |
| | μ (n = 1) | XP_036359829.1 *O. sinensis* | 47.11 | $1 \times 10^{-63}$ |
| | | XP_014772028.1 *O. bimaculoides* | 45.12 | $7 \times 10^{-63}$ |
| | | XP_029637902.1 *O. sinensis* | 44.75 | $1 \times 10^{-7}$ |
| | | XP_014772027.1 *O. bimaculoides* | 43.67 | $53 \times 10^{-76}$ |
| | S short loop (n = 3) | XP_036359020.1 *O. sinensis* | 49.46 | $4 \times 10^{-25}$ |
| | | XP_029636899.1 *O. sinensis* | 45.16 | $1 \times 10^{-22}$ |
| | | XP_036358738.1 *O. sinensis* | 42.86 | $1 \times 10^{-20}$ |
| | | XP_014783449.1 *O. bimaculoides* | 42.86 | $2 \times 10^{-20}$ |
| | | XP_036355061.1 *O. sinensis* | 37.40 | $4 \times 10^{-23}$ |
| | | XP_036356064.1 *O. sinensis* | 37.93 | $3 \times 10^{-27}$ |
| | ζ (n = 2) | XP_014777379.1 *O. bimaculoides* | 33.33 | $3 \times 10^{-09}$ |
| | | XP_029644700.1 *O. sinensis* | 32.52 | $1 \times 10^{-09}$ |
| | | XP_036359047.1 *O. sinensis* | 32.52 | $1 \times 10^{-09}$ |
| | | XP_014784767.1 *O. bimaculoides* | 31.28 | $5 \times 10^{-14}$ |
| *Pomacea canaliculata* | α (n = 8) | XP_029654771.1 *O. bimaculoides* | 45.00 | $5 \times 10^{-23}$ |
| | | XP_014783063.1 *O. sinensis* | 45.00 | $1 \times 10^{-22}$ |
| | | XP_014783064.1 *O. bimaculoides* | 44.14 | $5 \times 10^{-24}$ |
| | | XP_036354796.1 *O. sinensis* | 43.33 | $2 \times 10^{-24}$ |
| | | XP_029642860.1 *O. sinensis* | 41.67 | $7 \times 10^{-12}$ |
| | J1A (n = 1) | XP_014782921.1 *O. bimaculoides* | 47.35 | $1 \times 10^{-81}$ |
| | | XP_036358994.1 *O. sinensis* | 47.35 | $2 \times 10^{-81}$ |
| | λ (n = 3) | XP_036356168.1 *O. sinensis* | 47.12 | $2 \times 10^{-51}$ |
| | | XP_029654653.1 *O. sinensis* | 44.41 | $3 \times 10^{-80}$ |
| | | XP_014776757.1 *O. bimaculoides* | 44.41 | $2 \times 10^{-79}$ |
| | | XP_036356167.1 *O. sinensis* | 43.56 | $5 \times 10^{-75}$ |
| | μ (n = 2) | XP_014772028.1 *O. bimaculoides* | 46.56 | $3 \times 10^{-62}$ |
| | | XP_036359829.1 *O. sinensis* | 46.56 | $1 \times 10^{-60}$ |
| | | XP_014772027.1 *O. bimaculoides* | 44.44 | $1 \times 10^{-78}$ |
| | | XP_029637902.1 *O. sinensis* | 44.12 | $4 \times 10^{-76}$ |
| | S long loop (n = 1) | XP_036355061.1 *O. sinensis* | 39.78 | $6 \times 10^{-17}$ |
| | | XP_036359020.1 *O. sinensis* | 38.00 | $4 \times 10^{-17}$ |
| | | XP_029637027.1 *O. sinensis* | 37.75 | $2 \times 10^{-36}$ |
| | | XP_029636899.1 *O. sinensis* | 36.73 | $3 \times 10^{-16}$ |
| | | XP_029636493.1 *O. sinensis* | 34.95 | $5 \times 10^{-33}$ |

| Gastropoda molluscs | Crystallin | Mollusc Protein, GenBank Accession Number | Percent Identity, % | E-Value |
|---|---|---|---|---|
| | | versus *Bivalvia* | | |
| *Aplysia californica* | α (n = 1) | XP_045176285.1 *M. mercenaria* | 55.68 | $1 \times 10^{-54}$ |
| | | XP_045216391.1 *M. mercenaria* | 49.14 | $9 \times 10^{-46}$ |
| | | XP_033752285.1 *P. maximus* | 48.80 | $5 \times 10^{-43}$ |
| | | XP_045216364.1 *M. mercenaria* | 48.57 | $1 \times 10^{-44}$ |
| | | XP_021346583.1 *M. yessoensis* | 47.59 | $6 \times 10^{-41}$ |
| | J1A (n = 2) | XP_021365545.1 *M. yessoensis* | 51.95 | $4 \times 10^{-80}$ |
| | | XP_033731119.1 *P. maximus* | 51.29 | $3 \times 10^{-81}$ |
| | | XP_045200778.1 *M. mercenaria* | 49.21 | $1 \times 10^{-77}$ |
| | | XP_048774863.1 *O. edulis* | 44.87 | $6 \times 10^{-67}$ |
| | | XP_033743114.1 *P. maximus* | 42.86 | $1 \times 10^{-44}$ |
| | λ (n = 7) | XP_033739698.1 *P. maximus* | 62.34 | $7 \times 10^{-138}$ |
| | | XP_011439697.2 *C. gigas* | 62.21 | $1 \times 10^{-135}$ |
| | | XP_022295241.1 *C. virginica* | 61.87 | $9 \times 10^{-134}$ |
| | | XP_033739699.1 *P. maximus* | 61.84 | $2 \times 10^{-126}$ |
| | | XP_048771868.1 *O. edulis* | 61.20 | $3 \times 10^{-139}$ |
| | | XP_048762137.1 *O. edulis* | 57.49 | $1 \times 10^{-97}$ |
| | | XP_011454698.2 *C. gigas* | 54.64 | $4 \times 10^{-116}$ |
| | μ (n = 1) | XP_021361605.1 *M. yessoensis* | 53.02 | $3 \times 10^{-108}$ |
| | | XP_048762136.1 *O. edulis* | 52.82 | $8 \times 10^{-111}$ |
| | | XP_033746700.1 *P. maximus* | 52.68 | $2 \times 10^{-107}$ |
| | S short loop (n = 6) | XP_045194312.1 *M. mercenaria* | 50.00 | $5 \times 10^{-17}$ |
| | | XP_021339566.1 *M. yessoensis* | 45.00 | $1 \times 10^{-13}$ |
| | | XP_021378340.1 *M. yessoensis* | 41.76 | $2 \times 10^{-20}$ |
| | | XP_048759013.1 *O. edulis* | 40.30 | $3 \times 10^{-41}$ |
| | | XP_045167239.1 *M. mercenaria* | 39.71 | $4 \times 10^{-09}$ |
| | | XP_033754085.1 *P. maximus* | 36.82 | $4 \times 10^{-33}$ |
| *Biomphalaria glabrata* | α (n = 6) | XP_045166562.1 *M. mercenaria* | 42.27 | $3 \times 10^{-22}$ |
| | | XP_045202791.1 *M. mercenaria* | 39.33 | $6 \times 10^{-17}$ |
| | | XP_045176285.1 *M. mercenaria* | 38.71 | $1 \times 10^{-14}$ |
| | | XP_045203286.1 *M. mercenaria* | 38.64 | $2 \times 10^{-15}$ |
| | | XP_045192081.1 *M. mercenaria* | 38.64 | $4 \times 10^{-16}$ |
| | | XP_033730743.1 *P. maximus* | 34.41 | $2 \times 10^{-15}$ |
| | J1A (n = 2) | XP_021365545.1 *M. yessoensis* | 48.00 | $4 \times 10^{-36}$ |
| | | XP_033731119.1 *P. maximus* | 46.97 | $7 \times 10^{-38}$ |
| | | XP_045200778.1 *M. mercenaria* | 43.18 | $4 \times 10^{-35}$ |
| | | XP_022323163.1 *C. virginica* | 37.86 | $5 \times 10^{-23}$ |
| | | XP_011426564.1 *C. gigas* | 36.43 | $3 \times 10^{-20}$ |
| | λ (n = 2) | XP_048770804.1 *O. edulis* | 58.65 | $1 \times 10^{-134}$ |
| | | XP_034331263.1 *C. gigas* | 57.51 | $2 \times 10^{-132}$ |
| | | XP_034331265.1 *C. gigas* | 57.51 | $3 \times 10^{-132}$ |
| | | XP_045195917.1 *M. mercenaria* | 54.95 | $9 \times 10^{-131}$ |
| | | XP_034331611.1 *C. gigas* | 54.95 | $1 \times 10^{-119}$ |
| | | XP_033739696.1 *P. maximus* | 51.94 | $4 \times 10^{-119}$ |
| | μ (n = 1) | XP_011454698.2 *C. gigas* | 49.49 | $8 \times 10^{-89}$ |
| | | XP_021361605.1 *M. yessoensis* | 49.32 | $1 \times 10^{-86}$ |
| | | XP_048762137.1 *O. edulis* | 49.19 | $4 \times 10^{-71}$ |
| | | XP_033746700.1 *P. maximus* | 49.15 | $3 \times 10^{-87}$ |
| | | XP_022319189.1 *C. virginica* | 48.12 | $4 \times 10^{-85}$ |

Table 2. *Cont.*

| Gastropoda molluscs | Crystallin | Mollusc Protein, GenBank Accession Number | Percent Identity, % | E-Value |
|---|---|---|---|---|
| | S short loop (n = 3) | XP_045194312.1 *M. mercenaria* | 54.17 | $5 \times 10^{-21}$ |
| | | XP_034305106.1 *C. gigas* | 44.95 | $6 \times 10^{-25}$ |
| | | XP_021378340.1 *M. yessoensis* | 43.75 | $6 \times 10^{-23}$ |
| | | XP_033755358.1 *P. maximus* | 41.61 | $6 \times 10^{-36}$ |
| | | XP_034308304.1 *C. gigas* | 40.70 | $2 \times 10^{-42}$ |
| | ζ (n = 2) | XP_045204266.1 *M. mercenaria* | 52.16 | $1 \times 10^{-88}$ |
| | | XP_021380229.1 *M. yessoensis* | 51.88 | $2 \times 10^{-86}$ |
| | | XP_045204273.1 *M. mercenaria* | 51.84 | $7 \times 10^{-88}$ |
| | | XP_045204248.1 *M. mercenaria* | 51.84 | $1 \times 10^{-87}$ |
| | | XP_045204257.1 *M. mercenaria* | 50.92 | $2 \times 10^{-82}$ |
| | | XP_033732342.1 *P. maximus* | 49.38 | $1 \times 10^{-82}$ |
| *Pomacea canaliculata* | α (n = 8) | XP_045166562.1 *M. mercenaria* | 43.43 | $1 \times 10^{-20}$ |
| | | XP_045202791.1 *M. mercenaria* | 42.11 | $6 \times 10^{-19}$ |
| | | XP_045206475.1 *M. mercenaria* | 41.76 | $1 \times 10^{-13}$ |
| | | XP_045203286.1 *M. mercenaria* | 41.05 | $3 \times 10^{-18}$ |
| | | XP_022298136.1 *C. virginica* | 40.70 | $1 \times 10^{-13}$ |
| | | XP_033752285.1 *P. maximus* | 38.20 | $2 \times 10^{-11}$ |
| | J1A (n = 1) | XP_045200778.1 *M. mercenaria* | 56.15 | $8 \times 10^{-84}$ |
| | | XP_021365545.1 *M. yessoensis* | 56.13 | $8 \times 10^{-92}$ |
| | | XP_033731119.1 *P. maximus* | 55.84 | $7 \times 10^{-92}$ |
| | | XP_011426564.1 *C. gigas* | 50.96 | $1 \times 10^{-68}$ |
| | | XP_048774863.1 *O. edulis* | 50.57 | $5 \times 10^{-79}$ |
| | λ (n = 3) | XP_048770804.1 *O. edulis* | 46.73 | $2 \times 10^{-82}$ |
| | | XP_034331265.1 *C. gigas* | 46.23 | $2 \times 10^{-79}$ |
| | | XP_034331271.1 *C. gigas* | 45.63 | $4 \times 10^{-21}$ |
| | | XP_045195917.1 *M. mercenaria* | 45.62 | $3 \times 10^{-85}$ |
| | | XP_034331263.1 *C. gigas* | 45.22 | $6 \times 10^{-78}$ |
| | | XP_033739696.1 *P. maximus* | 43.91 | $6 \times 10^{-72}$ |
| | μ (n = 2) | XP_048762137.1 *O. edulis* | 54.88 | $3 \times 10^{-84}$ |
| | | XP_033746700.1 *P. maximus* | 53.02 | $2 \times 10^{-101}$ |
| | | XP_011454698.2 *C. gigas* | 53.00 | $1 \times 10^{-104}$ |
| | | XP_048762136.1 *O. edulis* | 52.32 | $2 \times 10^{-101}$ |
| | | XP_021361605.1 *M. yessoensis* | 52.16 | $2 \times 10^{-102}$ |
| | S long loop (n = 1) | XP_045194312.1 *M. mercenaria* | 44.74 | $1 \times 10^{-16}$ |
| | | XP_021378342.1 *M. yessoensis* | 42.36 | $7 \times 10^{-43}$ |
| | | XP_045164188.1 *M. mercenaria* | 42.05 | $3 \times 10^{-14}$ |
| | | XP_021378354.1 *M. yessoensis* | 41.87 | $1 \times 10^{-47}$ |
| | | XP_034305103.1 *C. gigas* | 41.67 | $4 \times 10^{-40}$ |
| | | XP_033755358.1 *P. maximus* | 38.68 | $6 \times 10^{-36}$ |

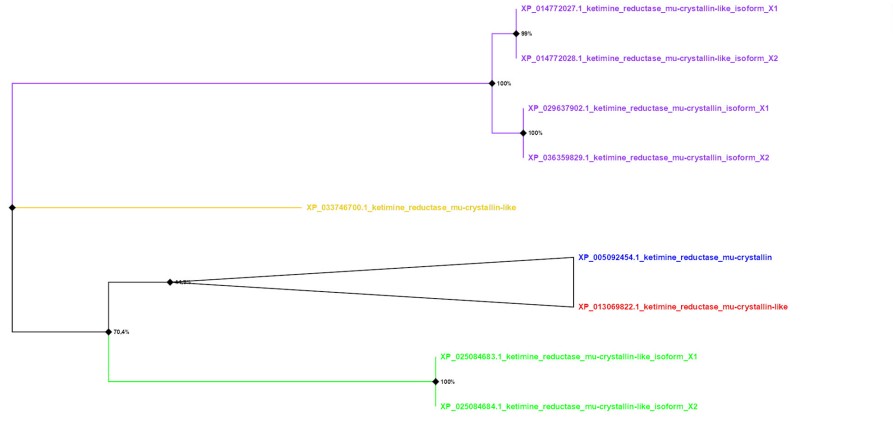

**Figure 8.** Phylogenetic tree of μ-crystallins of molluscs.

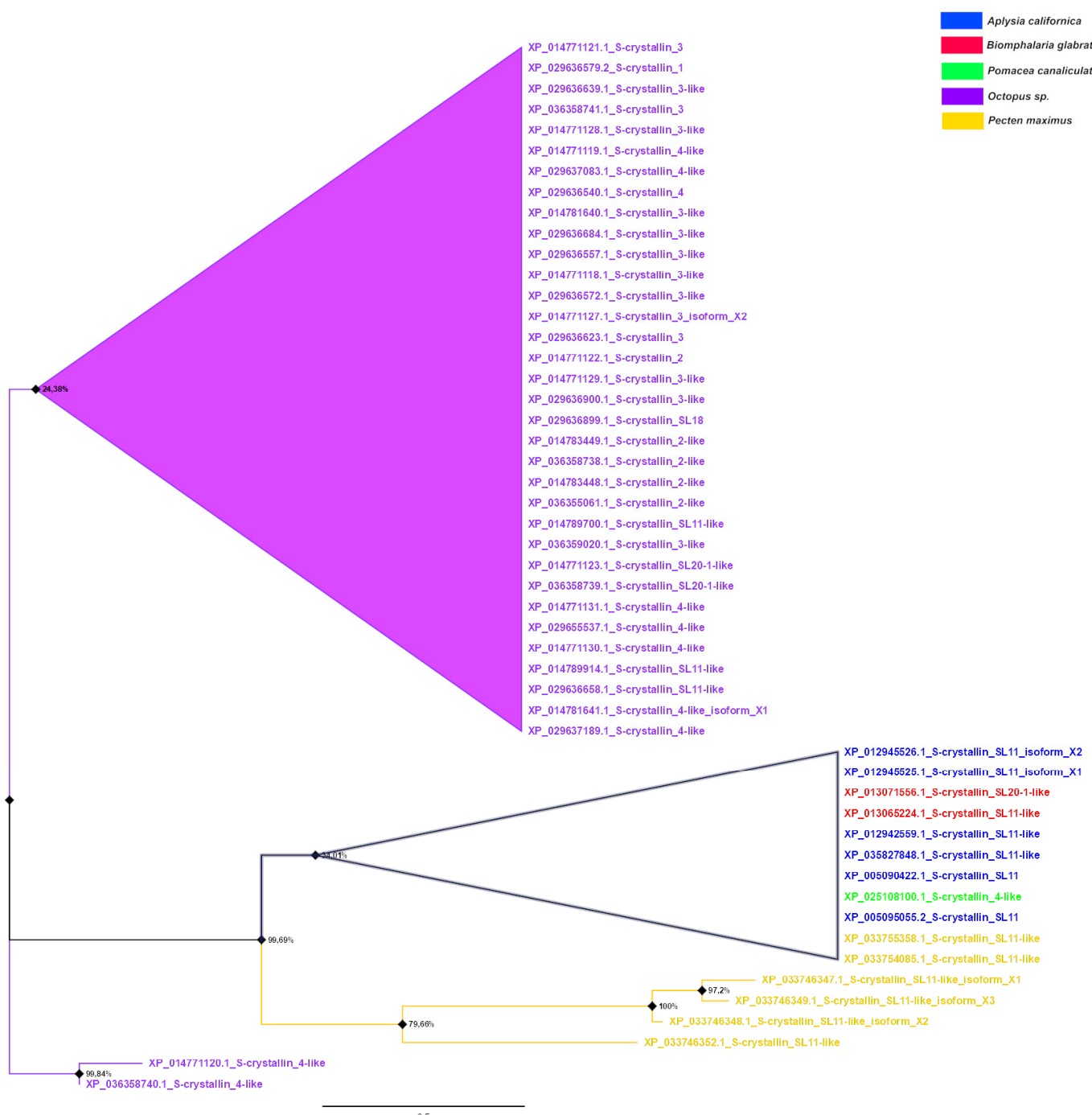

**Figure 9.** Phylogenetic tree of S-crystallins of molluscs.

The table shows the five proteins (where possible) with the highest percentage of identity, as well as the proteins of the organisms analyzed in this work if the proteins of these organisms were not in the first five proteins with the highest percentage of identity.

According to the results of identity determination using BLAST (Table 2), we can see that the gastropod crystallins did not have high similarity with crystallins of cephalopods and bivalve molluscs. A maximum of about 62% identity was observed only for J1A-crystallins of Aplysia compared to J1A-crystallins of bivalves.

## 4. Discussion

Crystallins constitute a polyfunctional group of proteins found in various organs and tissues. The most obvious biological role of crystallins is in the lens, where they provide stability and transparency of its structure, as well as creating refractive power corresponding to the animal's habitat [39]. The latter is of particular importance for the chamber eyes of aquatic organisms, in which it is the lens that is the main refractive element of the optical system of the eye [40]. However, data on the crystalline content in the lens of aquatic animals have been obtained only for a few such animals, e.g., the fish *D. rerio* [35], jellyfish *T. cystophora* [15], bivalve mollusc *P. maximus* [41,42], and cephalopods, namely squids of genus *Doryteuthis* [43,44] and *Octopus vulgaris* [45]. As for the gastropod molluscs, it is unknown which of the crystallins identified in them may be part of the lens. Therefore, it seems reasonable to consider the features of the structural and optical properties of lens crystallins of the above aquatic organisms in comparison with similar proteins of gastropod molluscs to assess the probability of their providing lens function. Quantitative values of all analyzed parameters are shown in Table 3.

**Table 3.** Main structural and physicochemical parameters of known mollusc crystallins.

| Organism | Crystallin | GRAVY | Prevailing Secondary Structure, % | Proportion of 7 Amino Acids with the Highest Value of the Increment of Refractive Index, % | Calculated Average Increment of Refractive Index Values ($dn/dc$, mL/g) |
|---|---|---|---|---|---|
| *Octopus* sp. | α (n = 5) | −0.7226 ± 0.1231 | 63.20% ± 6.54 Coil | 23.00% ± 3.23 | 0.1885 ± 0.0018 |
| | J1A (n = 3) | −0.2980 ± 0.0225 | 57.00% Alpha Helix | 15.23% ± 0.46 | 0.1857 ± 0.0006 |
| | λ (n = 4) | −0.1243 ±0.0899 | 49.75% ± 6.75 Alpha Helix | 17.15% ± 0.56 | 0.1854 ± 0.0003 |
| | μ (n = 4) | 0.1430 ± 0.0530 | 41.00% ± 2.71 Coil | 15.68% ± 0.78 | 0.1840 ± 0.0002 |
| | Ω (n = 2) | −0.3710; −0.3780 | 43.00% Coil | 21.20%; 21.60% | 0.1878 |
| | S long loop [1] (n = 21) | −0.6250 ± 0.1360 | 53.62% ± 4.40 Alpha Helix | 39.45% ± 2.91 | 0.1938 ± 0.0009 |
| | S short loop (n = 18) | −0.5134 ± 0.1373 | 54.78% ± 4.71 Alpha Helix | 31.79% ± 5.36 | 0.1917 ± 0.0018 |
| | ζ (n = 1) | 0.1950 | 44.00% Coil | 20.30% | 0.1861 |
| *Pecten maximus /Placopecten magellanicus* | α (n = 3) | −0.6657 ± 0.3234 | 61.67% ± 1.53 Coil | 24.97% ± 3.09 | 0.1882 ± 0.0014 |
| | J1A (n = 3) | −0.2590 ±0.0641 | 56.67% ± 1.16 Alpha Helix | 18.00% ± 0.50 | 0.1855 ± 0.0005 |
| | λ (n = 4) | −0.2590 ± 0.0786 | 51.25% ± 2.22 Alpha Helix | 19.55% ± 0.31 | 0.1861 ± 0.0002 |
| | μ (n = 1) | 0.0880 | 43.00% Coil | 12.60% | 0.1825 |
| | Ω [1] (n = 1) | −0.1080 | 42.00% Coil | 18.00% | 0.1859 |
| | S short loop (n = 6) | −0.1713 ± 0.3945 | 53.33% ± 3.39 Alpha Helix | 19.45% ± 3.12 | 0.1869 ± 0.0016 |
| *Aplysia californica* | α (n = 1) | −0.6620 | 60.00% Coil | 24.30% | 0.1883 |
| | J1A (n = 2) | −0.2460; −0.2870 | 51.00% & 60.00% Alpha Helix | 16.80%; 17.40 | 0.1847 |
| | λ (n = 7) | −0.1139 ± 0.0428 | 51.86% ± 0.90 Alpha Helix | 18.17% ± 0.92 | 0.1850 ± 0.0009 |
| | μ (n = 1) | −0.0060 | 44.00% Coil | 16.30% | 0.1837 |
| | S short loop (n = 6) | −0.2362 ± 0.0807 | 53.00% ± 1.79 Alpha Helix | 20.72% ± 2.38 | 0.1881 ± 0.0016 |
| *Biomphalaria glabrata* | α (n = 6) | −0.7657 ± 0.1068 | 53.17% ± 3.82 Coil | 16.48% ± 2.79 | 0.1852 ±0.0010 |
| | J1A (n = 2) | −0.2480; −0.4090 | 56.00% Alpha Helix; 55.00% Coil | 21.00%; 23.00% | 0.1866; 0.1894 |
| | λ (n = 2) | −0.1470; −0.0970 | 51.00% Alpha Helix | 18.10%; 19.00% | 0.1848; 0.1859 |
| | μ (n = 1) | 0.2200 | 42.00% Coil | 16.60% | 0.1844 |
| | S short loop (n = 3) | −0.1920 ± 0.0439 | 59.67% ± 11.59 Alpha Helix | 20.67% ± 1.55 | 0.1882 ± 0.0006 |
| | ζ (n = 2) | −0.0690; 0.2060 | 44.00%; 48.00% Coil | 13.00%; 17.20% | 0.1830; 0.1850 |
| *Pomacea canaliculata* | α (n = 8) | −0.6266 ± 0.1022 | 60.6250% ± 6.3230 Coil | 21.96% ± 1.60 | 0.1873 ± 0.0008 |
| | J1A (n = 1) | −0.3640 | 56.00% Alpha Helix | 18.10% | 0.1858 |
| | λ (n = 3) | −0.1807 ± 0.0560 | 51.33% ± 1.53 Alpha Helix | 20.10% ± 2.17 | 0.1855 ± 0.0006 |
| | μ (n = 2) | 0.1130; 0.0480 | 43.00%; 45.00% Coil | 14.90%; 14.80% | 0.1832 |
| | S long loop (n = 1) | −0.5500 | 57.00% Alpha Helix | 23.50% | 0.1903 |

[1] Confirmed lens crystallins. All data are presented as mean ± SD if n ≥ 2.

The overall values of the GRAVY index in general showed rather high hydrophilicity (<−50.0) of the lens crystallin molecules in comparison with ones for which belonging to lens has not been confirmed. Thus, all the crystallins of *D. rerio* lens, the jellyfish *Tripedalia*, and S-crystallins of the octopus had a high hydrophilicity. Hydrophilicity was also strongly expressed in the J-crystallins of the lens of the cube jellyfish *Tripedalia* (Figure 3i), and slightly less so in the octopus (Figure 3b). The small value of hydrophilicity (−0.1080) of Ω-crystallins of *Pecten* (Figure 3e) may be related to the specificity of the scallop eye optical system, in which the main focusing element is not the lens, but the reflective

layer underlying the retina [46]. The degree of hydrophilicity of α-crystallins in all molluscs studied in this regard was well expressed, approaching the main crystallins of the *D. rerio* lens: α, β, γM, γN, γS (Figure 3h). Moreover, in octopus, the hydrophilicity of α-crystallins even exceeded that of the S-crystallins, the main proteins of its lens. Assuming that crystallins of the lens should have a pronounced hydrophilicity, ζ-, µ-, and λ-crystallins should be excluded from the list of candidates for gastropod lens crystallins.

The assessment results of the secondary structure of crystallins look rather contradictory. Thus, β-strands predominated in all *D. rerio* lens crystallins, reaching 40% for proteins of the γM group (Figure 4h), which is generally consistent with the ideas about the necessity of dense protein packing in the lens to achieve high values of refractive power and a radial gradient of the refractive index. In *D. rerio*, this index varied from 1.57 to 1.58 in the center of the lens to 1.35 in the peripheral region [47]. However, in the structures of S-crystallins of cephalopods, the proportion of α-helices exceeded 50% but the proportion of β-strand regions barely reached 10%. Moreover, this was true for crystallins both with a long loop (specialized lens proteins) and a shortened loop. The proportion of α-helices was also relatively high (more than 25%) in J1-crystallins of the jellyfish *Tripedalia* lens. At the same time, although the refractive index of the substance of the central region of the lens of the jellyfish was noticeably higher than that of its edge (1.47–1.48 vs. 1.33) [48], it did not reach such values as in *Danio* and cephalopods (1.509 in the octopus and 1.62 in the squid). The proportion of α-helices (about 38%) of molecules was also high in Ω-crystallins in cephalopods and scallops. However, these proteins are not the main component of the octopus' lens, and the refractive power of the scallop eye lens is low. The α-helix fraction in α-crystallins in all molluscs was low (about 10%), bringing them closer to the main proteins of the lens of *D. rerio* [36].

All *D. rerio* γ-crystallins led in the value of the increment of refractive index (*dn/dc*) (≥0.195). This parameter was somewhat less, but still higher than 0.190, for β-crystallins. In the octopus, long-loop S-crystallins also had a high value of this index, about 0.1938. These values were noticeably higher than for other crystallins, which probably explains the role of these proteins in creating a radial refractive index gradient and allows correction of the spherical aberration of the spherical lens in *Danio* and cephalopods [36]. The spherical lens of the jellyfish *T. cystophora* [47] has the same optical property, for which J1-crystallin is probably responsible. The refractive properties of Ω- and S-crystallins of the scallop lens were not strongly pronounced, and in *P. maximus* they were even less than that of α-crystallin, but this is consistent with the above features of the eye optical systems of these molluscs. As for gastropods, α- and S-crystallins had the highest *dn/dc* values in *A. californica*, J1A and S-crystallins in *B. glabrata*, and the only S-crystallin in *P. caniculata*. Direct measurements of the refractive index values of various regions of the lens of the gastropods mentioned in the article were not carried out. The values of the refractive index of aquatic gastropods measured by the crystallographic method were rather high, close to those of fish: *Lymnaea stagnalis*—1.545–1.560 [49], *Viviparus viviparous*—1.550–1.570 [50]. The presence of the radial gradient of the lens refractive index is indicated by the value of its focal distance, close to the Matthysson radius of the fish eye lens, about 2.5R. For example, *Littorina littorea*—2.3–2.8 [51,52], *Strombus raninus*—approximately 2.0 [53], *Viviparus viviparous*—2.44 [49], *Littorina irrorata*—2.71 [54], *Lymnaea stagnalis*, *Physa fontinalis*, and *Radix peregra*—2.70–2.90 [48].

Since the *dn/dc* value is calculated from the corresponding indices of individual amino acids and their share in the molecule structure, it is not surprising that the obtained values correlated with the share of the most polarizable amino acids in the crystallin composition. Note, however, that the relative content of these amino acids varies in different hydrobionts, including in crystallin protein molecules and those that can be proposed for this role.

According to the combination of characteristics, proteins of groups α, S, and maybe also J1A can be considered the most likely candidates for the role of lens crystallins of gastropod molluscs (in this study *A. californica*, *P. canaliculata*, *B. glabrata*).

Today, the most widely accepted concept is that the *Mollusca* taxon is divided into *Aculifera* and *Conchifera* (Figure 10) [55,56]. However, it should be noted that, apart from the hypothesis under discussion concerning the phylogenetic relationships between classes of Mollusca, there are several other hypotheses [57]. At the same time, the studied mollusc groups are in different clades: *Bivalvia* and *Gastropoda* are combined into one, and *Cephalopoda* into another. We saw this separation to the greatest extent in the phylogenetic trees of S- and μ-crystallins: proteins of cephalopods were located on branches that did not overlap with those of bivalves and gastropods. To a somewhat lesser extent, a similar divergence was traced for J1A-crystallins.

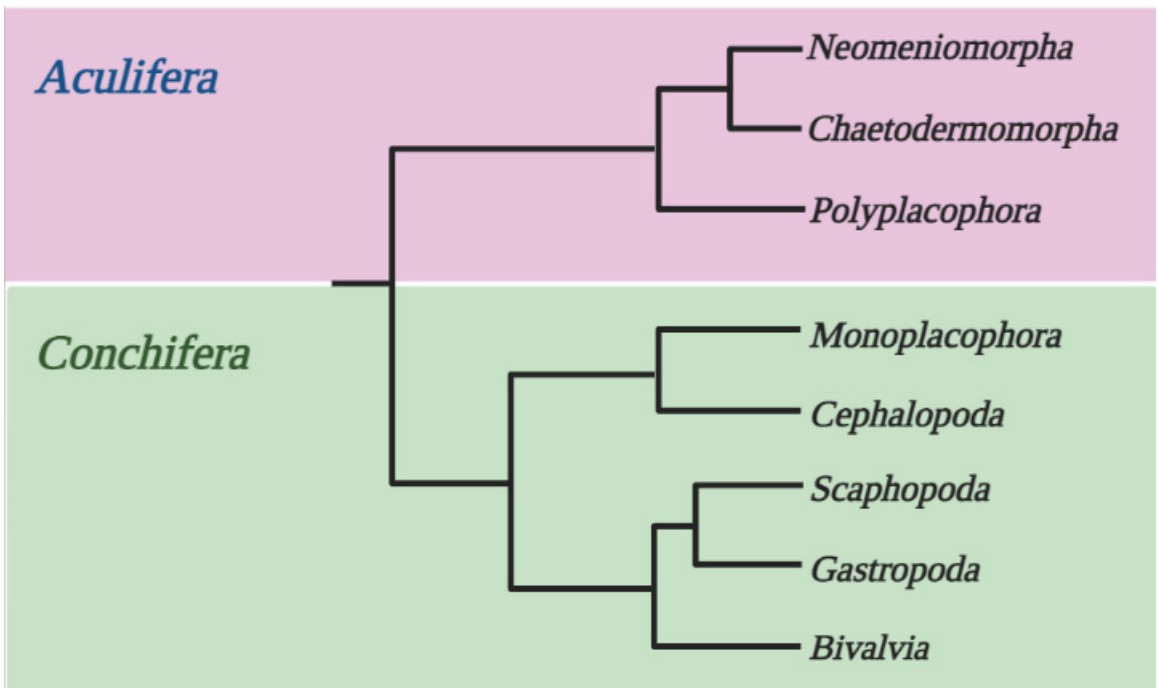

**Figure 10.** Current understanding of Mollusca phylum taxonomy (adapted from [56] by BioRender.com (2022)).

The phylogenetic trees of α- and λ-crystallins look significantly different: the proteins belonging to different groups of molluscs are distributed in a more fragmentary way. Based on the fact that S-crystallins predominate in octopus' lenses, the isolation of the corresponding branch of the phylogenetic tree can be considered an indication of the specialization of these proteins. The same can be said about J1A-crystallins. As for the phylogenetic separation of μ-crystallins, their specialization may not be related to their possible presence in the lens, which has not been confirmed, especially considering their pronounced hydrophobicity. It should be noted that the low bootstrap values on some tree branches can be explained by two factors. First, the study was performed on non-model animals for which there are no validated gene sequences and, consequently, proteins expressed from them. All sequences were predicted using bioinformatic approaches, particularly by Gnomon. Of course, the analyzed sequences require validation, which may constitute the subject of further research. Second, due to the insufficient study of these organisms, we cannot exclude the high variability of the amino acid sequences. Therefore, we could not unambiguously determine which of the selected sequences are "wrong" in order to remove them from the analysis. Moreover, according to BLAST analysis (Table 2), the percentage of identity between the amino acid sequences of crystallins of gastropods and cephalopods and bivalves was quite low (for the most part no more than 50%), but the e-value values for all proteins, except the ζ-crystallins of *B. glabrata* compared with *O. sinesis*, were lower than $10^{-09}$, indicating the presence of conserved sequences with a

sufficiently high level of homology in the analyzed sequences of gastropods, with similar sequences in cephalopods and gastropod molluscs. At the same time, the low percentage of identity shows the presence of a large number of amino acid positions in the crystallins of gastropod molluscs that do not align with the sequences of cephalopods and gastropods, which may be evidence of strong evolutionary divergence of representatives of this class. However, the limitations discussed above also restrict the conclusions that can be drawn about the evolution of the proteins in question. Furthermore, the low percentage of identity may be due to the lack of validated crystallin sequences for the molluscs, for example using transcriptomic or proteomic analyses. Thus, we return again to the fact that molluscs are still a largely unexplored taxonomic group of animals and require further study.

In predicting crystallins for the role of lens proteins, one should keep in mind the different degrees of structural organization of the optical apparatus and retina. The camera-type eye of gastropods can be referred to a group of as less complexity compared to the fundamentally similar-in-structure eyes of cephalopods and vertebrates, including fishes. It is quite possible that the evolution of lens crystallins at this stage has not yet led to the formation of these lens proteins into an independent group. In this respect, a parallel can be drawn with one of the proteins of the visual cycle, arrestin. This protein has undergone phylogenetic specialization from β-arrestin, a widespread molecular component of GPCR signal cascades, to the visual arrestin of the cephalopod retina [58]. Using the slug *Limax valentianus* as an example, it was shown that the same form of β-arrestin functions in the photoreceptors of the simpler gastropod eye and nonphotoreceptor cells [59]. It is possible that α-crystallins common in the body of gastropod molluscs are not yet specialized, and perform their role in the lens in addition to their other functions in the other cells of the organism. Meanwhile, S-crystallins of cephalopod lenses have undergone the phylogenetic path of specialization from glutathione-S-transferase to proteins of the optical apparatus in the developed eye [6]. However, it is obvious that the assumption about the phylogenetic incompleteness of the specialization of gastropod eye crystallins is rather arbitrary and can be verified only by detailed studies of these proteins. The currently available information is still quite insufficient for more definite conclusions.

The results proposed in this paper are preliminary and based solely on bioinformatic data, and require verification based on transcriptomic and proteomic analysis.

**Supplementary Materials:** The following supporting information can be downloaded at: https://www.mdpi.com/article/10.3390/d14100827/s1, Figure S1: Alignment of mollusc crystallin amino acid sequences; Table S1: Crystallin amino acid sequences; Table S2: Analysis of multiple sequence alignment of crystallins and crystallin-like proteins of molluscs.

**Author Contributions:** Conceptualization, I.N.D. and V.V.Z.; methodology, I.N.D.; software, I.N.D.; validation, I.N.D.; formal analysis, I.N.D.; investigation, I.N.D.; resources, I.N.D.; data curation, I.N.D.; writing—original draft preparation, I.N.D. and V.V.Z.; writing—review and editing, I.N.D. and V.V.Z.; visualization, I.N.D.; supervision, V.V.Z.; project administration, V.V.Z. All authors have read and agreed to the published version of the manuscript.

**Funding:** This research received no external funding.

**Institutional Review Board Statement:** Not applicable.

**Informed Consent Statement:** Not applicable.

**Data Availability Statement:** Publicly available datasets were analyzed in this study. These data can be found here: https://www.ncbi.nlm.nih.gov/ (accessed on 15 July 2022).

**Conflicts of Interest:** The authors declare no conflict of interest.

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
