# Peer review of "Mollusc Crystallins: Physical and Chemical Properties and Phylogenetic Analysis"

_diversity, doi:10.3390/d14100827_

Round 1

Reviewer 1 Report

It is an interesting study which would attract attention of experts in many fields, ranging from “pure” malacology to neurophysiology. Though the overall quality of this manuscript is high, I recommend to revise it in accordance with the below recommendations.

Lines 62-64 “The only one work reported about the isolation of three polypeptides (80, 63, and 28 kDa), presumably crystallins, from the marine mollusc Aplysia californica (J.G. Cooper, 1863) [12,13] was not further investigated”. The meaning of this phrase is not clear to me. Could you please reword?

Line 70 “hydrobiont molluscs” – I recommend to use the term “aquatic molluscs”; “hydrobiont molluscs” sounds awkward.

Line 100. For ProtParam software and other software products used in the study, the name and nationaliy of their producers must be indicated. If these software products are in free access, please provide the URLs for their sites.

Line 305 “Phylogenetic tree of ɑ-crystallins of molluscs”. I am not sure that the term “phylogeny” is applicable in this context. Phylogeny is applied to whole organisms (or clades) not to proteins. Is this tree showing the overall similarity among studied proteins, rather than their alleged “phylogeny”? This comment is applied also to Fig. 6-9. Furthermore, many clades on these figures have very low bootstrap values. I wish to ask the authors, whether such weakly supported clades are worthy to discuss at all? This fact must, at least, be acknowledged and explained in the ‘Discussion’ chapter.

Line 318 “They form two formed clades”. That sounds tautologically. Please consider a rewording.

Lines 478-484. Though the authors discuss only one hypothesis concerning the deep phylogenetic relationships among classes of Mollusca, there are several alternative hypotheses. Moreover, the true phylogenetic relationships among the classes remain unresolved and there is a suspection that this will be a persistent problem for a long time (see Schroedl, M. & Stцger, I. A review on deep molluscan phylogeny: old markers, integrative approaches, persistent problems. J. Nat. Hist. 48, 2773–2804 (2014).

Fig. 10 is based on a source published in 2014, i.e., 8 years ago. A short term, indeed, but the field is developing fastly, and more recent phylogenetic hypotheses may be cited in this context. See, for example, Kocot et al. (2020) study - https://doi.org/10.1038/s41598-019-56728-w

I feel that for virtually each “phylogenetic” tree presented on figs. 6-9 a suitable phylogenetic hypothesis published by a certain team of malacologists can be found.

Fig. 10 Scapholopoda is an error. The correct name of the class is Scaphopoda. Please correct this.

I have made a series of minor changes and suggestions in the pdf-version of the manuscript.

Author Response

Dear Reviewer,

Thank you very much for your comments, concerns and suggestions We think that your input is very important, and we hope we could answer all questions, and provide you valuable informative answers. We also corrected all minor issues and typos in the text. We would like to note that all data will be freely available online in case the manuscript will be accepted for publication.

Thank you for your time and patience,

Best regards,

Irina Dominova.

Bellow we provide the answers for all questions.

  1. Line 62-64 “The only one work reported about the isolation of three polypeptides (80, 63, and 28 kDa), presumably crystallins, from the marine mollusc Aplysia californica (J.G. Cooper, 1863) [12,13] was not further investigated”.

The following sentences were added to the text:

The only one work reported about the isolation of three polypeptides (80, 63, and 28 kDa), presumably crystallins, from the marine mollusc Aplysia californica (J.G. Cooper, 1863). To date, however, this research has not been further developed and has remained incomplete.  

  1. Line 70 “hydrobiont molluscs” – I recommend to use the term “aquatic molluscs”; “hydrobiont molluscs” sounds awkward.

Corrected

  1. Line 100. For ProtParam software and other software products used in the study, the name and nationaliy of their producers must be indicated. If these software products are in free access, please provide the URLs for their sites.

Corrected

  1. Line 305 Phylogenetic tree of ɑ-crystallins of molluscs”. I am not sure that the term “phylogeny” is applicable in this context. Phylogeny is applied to whole organisms (or clades) not to proteins. Is this tree showing the overall similarity among studied proteins, rather than their alleged “phylogeny”? This comment is applied also to Fig. 6-9. Furthermore, many clades on these figures have very low bootstrap values. I wish to ask the authors, whether such weakly supported clades are worthy to discuss at all? This fact must, at least, be acknowledged and explained in the ‘Discussion’ chapter.

The term “Phylogenetic trees” is commonly used now for graphic representation of phylogenetic relationship between as whole organisms as well as genes or proteins (https://doi.org/10.1016/j.ygeno.2018.08.010).

The following fragment was added to the text:

It should de noted that the low bootstrap values on some tree branches can be explained by two factors. First, the study was performed on non-model animals for which there are no validated gene sequences and, consequently, proteins expressed from them. All sequences were predicted using bioinformatic approaches, for example by Gnomon. Second, due to the insufficient study of these organisms, we cannot exclude the high variability of the amino acid sequences. Therefore, we cannot unambiguously determine which of the selected sequences are "wrong" in order to remove them from the analysis. 

  1. Line 318 “They form two formed clades”. That sounds tautologically. Please consider a rewording.

Improved

  1. Lines 478-484. Though the authors discuss only one hypothesis concerning the deep phylogenetic relationships among classes of Mollusca, there are several alternative hypotheses. Moreover, the true phylogenetic relationships among the classes remain unresolved and there is a suspection that this will be a persistent problem for a long time (see Schroedl, M. & Stцger, I. A review on deep molluscan phylogeny: old markers, integrative approaches, persistent problems. Nat. Hist. 48, 2773–2804 (2014).

The following sentences were added to the text:

It should be noted that, apart from the hypothesis under discussion concerning the phylogenetic relationships between classes of Mollusca, there are several other hypotheses (for example, Schrödl, M. & Stöger, I. A review on deep molluscan phylogeny: old markers, integrative approaches, persistent problems. J. Nat. Hist. 2014. 48, 2773–2804).

  1. 10 is based on a source published in 2014, i.e., 8 years ago. A short term, indeed, but the field is developing fastly, and more recent phylogenetic hypotheses may be cited in this context. See, for example, Kocot et al. (2020) study - https://doi.org/10.1038/s41598-019-56728-w

 We apologize for the typo in the link. It has been corrected, the year of publication is 2018.

  1. I feel that for virtually each “phylogenetic” tree presented on figs. 6-9 a suitable phylogenetic hypothesis published by a certain team of malacologists can be found.

In the resulting trees, only octopus’s proteins stand out in separate clades. At the same time, the crystallins of the other molluscs are located more scattered.

In the phylogenetic diagrams, the mutual position of molluc taxa depends on the chosen method of tree construction (Kocot et al., 2020). Therefore, it does not seem to us very productive to select a certain scheme of phylogenetic relationships of organisms for each specific group of proteins.

  1. Fig. 10 Scapholopoda is an error. The correct name of the class is Scaphopoda. Please correct this

Corrected

Reviewer 2 Report

The manuscript by Dominova and Zhukov reports the analysis of the crystalline sequences identified in different molluscan species, with the aim to identify the most likely gastropod eye crystallins.

Not being an expert in structural biology, I will here provide my report mostly concerning the evolutionary side of this story.

The introduction is complete, as all necessary information has been provided and literature on the subject has been adequately cited. However, the authors might want to be more specific about bivalves, since the visual function has been exclusively studied (to the best of my knowledge) in scallops, i.e. Pectinida, which only comprise a small number of species compared to the high biodiversity found in this class of mollusks. Scallops  do indeed possess eyes, but many other bivalves do not have such sophisticated visual sensors.

L89: the authors should explain how crystalline sequences were identified: was a BLAST-based strategy used? Were the sequences screened looking for the presence of particular molecular signatures (e.g. the presence of conserved protein domains)? It would be also important to specify whether any threshold (of similarity or e-value) was used to define a sequence as belonging to a given crystalline family or not, as it would simplify the interpretation of section 3.

L130-L137: how was  the model of molecular evolution selected? Did the authors use modeltest or another similar algorithm to detect the best-fitting molecular model of evolution?

Figure 1 and following gigures: this will probably have only a negligible effect on the calculations, but are crystallins processed in any way during their maturation?  For example, it they are secretory proteins, the short signal peptide region at the N-terminus should be excluded from all calculations, which should only start from the first amino acid of the mature peptide.

L262: “medium conserved” is an arbitrary definition, which should be avoided.

L263: “domain” has quite a strict definition. Do crystallins possess conserved domains (if this is the case, which domains can be identified?) or do they rather show the presence of conserved motifs?

L268: First, it is pretty clear that the different sequences taken into account display, for most crystalline families, a highly variable length. This could either reflect a real diversity in sequence length or the inclusion of partial, truncated sequences, which could not be appropriately aligned with the others. Please note that this is a very common case in molluscan genomes (and more in general in all non-model species’ genomes), since gene annotation is an imperfect automated process, which relies on genomes assemblies of variable quality, on the detection of homology vs a given sequence database and/or on the availability of RNA-seq data. Such process often struggles with the annotation of genes located in genomic regions which have been poorly resolved, which include errors or which lack high homology with other sequences deposited in publicly available databases or  transcriptomic confirmation. Also, truncated pseudogenes may be occasionally annotated as protein-coding genes. In summary, the authors should definitely pay more attention to such cases, excluding obviously truncated and incomplete proteins, as they are unlikely to represent “real” sequences. This could, of course, impact their calculations and, above all, phylogenetic analyses.

By looking at figure S1, it is quite clear that the authors have only used a portion of the sequences to build multiple sequence alignments, for example by removing the N-terminal signal peptide (I think) and perhaps some C-terminal extensions. This is totally fine, as phylogenetically uninformative regions should be always removed, but this should be explained in the materials and methods section.

From the same figure, it is apparent that several “wrong” sequences have been included. This is not the fault of the authors, but as I said above gene predictions are often incorrect. In figure S1B for example, the two B. glabrata proteins are obviously both largely incomplete and combine a correct portion of sequence with a few random amino acids. Both sequences should be removed, in particular because these two sequences share no amino acids in the MSA and therefore estimating their pairwise distance would be simply not feasible (as a matter of fact, it looks like they may belong to the very same truncated gene).

Figure S1C shows that one of the Octopus sequences has an incorrect C-terminus. This is clearly visible, since the sequence suddenly starts to be extremely divergent from all the others, as the likely result of an incorrect exon prediction. The sequence could be kept (in general, the authors should arbitrarily choose to keep all sequences displaying a length > 50% of the full alignment), but the portion of this sequence which is obviously wrong should be cropped, as it would otherwise incorrectly lead to an over-estimate of the distance between this sequence and the others.

Figure S1F shows several such cases: remove incorrect sequence regions, keep only sequences alignable with each other and long enough to be informative.

In general, the authors should understand that reporting a range of sequence length in the table they display in the main text does not make much sense, considering the fact that several short sequences obviously derive from incorrect annotations (i.e. they are not real proteins). Al calculations and phylogenetic trees should be updated accordingly.

Graphical representation of the phylogenetic trees.  In general, the current representation is not clear, since the authors chose to report binary trees in all cases, regardless of the bootstrap support of the nodes, which was often very low, and collapse some portions of the tree in cartoons, even though this was unnecessary. First of all, the quality of a phylogenetic tree is based on the quality of the underlying multiple sequence alignment. As I mentioned above, it looks like the authors might have included several highly divergent, incomplete sequences in the alignment, which might have resulted in inconsistent analyses. All nodes showing poor boostrap support (i.e. < 50) should be collapsed and presented as multifurcate. Representing binary nodes with support values lower than that has no biological meaning and would be indeed quite misleading. The tree rooting strategy (midpoint? User-selected? Based on an outgroup?) should be also explicitly stated. Reporting branch lengths over each branch only adds confusion: the scale bar is sufficient.

Author Response

Dear Reviewer,

Thank you very much for your comments, concerns and suggestions We think that your input is very important, and we hope we could answer all questions, and provide you valuable informative answers. We also corrected all minor issues and typos in the text. We would like to note that all data will be freely available online in case the manuscript will be accepted for publication.

Thank you for your time and patience,

Best regards,

Irina Dominova.

Bellow we provide the answers for all questions.

  1. L89: the authors should explain how crystalline sequences were identified: was a BLAST-based strategy used? Were the sequences screened looking for the presence of particular molecular signatures (e.g. the presence of conserved protein domains)? It would be also important to specify whether any threshold (of similarity or e-value) was used to define a sequence as belonging to a given crystalline family or not, as it would simplify the interpretation of section 3.

Crystallin sequences were selected based on the available annotations in the NCBI Protein bank. Therefore, no e-value was available.

  1. L130-L137: how was  the model of molecular evolution selected? Did the authors use modeltest or another similar algorithm to detect the best-fitting molecular model of evolution?

Corrected. The model test was performed by IQ-TREE Web Server.

  1. Figure 1 and following figures: this will probably have only a negligible effect on the calculations, but are crystallins processed in any way during their maturation? For example, it they are secretory proteins, the short signal peptide region at the N-terminus should be excluded from all calculations, which should only start from the first amino acid of the mature peptide.

The study was performed on non-model animals for which there are no validated gene sequences and, consequently, proteins expressed from them. All sequences were predicted using bioinformatic approaches, for example by Gnomon.

  1. L262: “medium conserved” is an arbitrary definition, which should be avoided.

We change the term “medium conserved” to the “semi-conserved”.

  1. L263: “domain” has quite a strict definition. Do crystallins possess conserved domains (if this is the case, which domains can be identified?) or do they rather show the presence of conserved motifs?

We change the term “domain” to the “motif”.

  1. L268: First, it is pretty clear that the different sequences taken into account display, for most crystalline families, a highly variable length. This could either reflect a real diversity in sequence length or the inclusion of partial, truncated sequences, which could not be appropriately aligned with the others. Please note that this is a very common case in molluscan genomes (and more in general in all non-model species’ genomes), since gene annotation is an imperfect automated process, which relies on genomes assemblies of variable quality, on the detection of homology vs a given sequence database and/or on the availability of RNA-seq data. Such process often struggles with the annotation of genes located in genomic regions which have been poorly resolved, which include errors or which lack high homology with other sequences deposited in publicly available databases or transcriptomic confirmation. Also, truncated pseudogenes may be occasionally annotated as protein-coding genes. In summary, the authors should definitely pay more attention to such cases, excluding obviously truncated and incomplete proteins, as they are unlikely to represent “real” sequences. This could, of course, impact their calculations and, above all, phylogenetic analyses.

By looking at figure S1, it is quite clear that the authors have only used a portion of the sequences to build multiple sequence alignments, for example by removing the N-terminal signal peptide (I think) and perhaps some C-terminal extensions. This is totally fine, as phylogenetically uninformative regions should be always removed, but this should be explained in the materials and methods section.

From the same figure, it is apparent that several “wrong” sequences have been included. This is not the fault of the authors, but as I said above gene predictions are often incorrect. In figure S1B for example, the two B. glabrata proteins are obviously both largely incomplete and combine a correct portion of sequence with a few random amino acids. Both sequences should be removed, in particular because these two sequences share no amino acids in the MSA and therefore estimating their pairwise distance would be simply not feasible (as a matter of fact, it looks like they may belong to the very same truncated gene).

Figure S1C shows that one of the Octopus sequences has an incorrect C-terminus. This is clearly visible, since the sequence suddenly starts to be extremely divergent from all the others, as the likely result of an incorrect exon prediction. The sequence could be kept (in general, the authors should arbitrarily choose to keep all sequences displaying a length > 50% of the full alignment), but the portion of this sequence which is obviously wrong should be cropped, as it would otherwise incorrectly lead to an over-estimate of the distance between this sequence and the others.

Figure S1F shows several such cases: remove incorrect sequence regions, keep only sequences alignable with each other and long enough to be informative.

In general, the authors should understand that reporting a range of sequence length in the table they display in the main text does not make much sense, considering the fact that several short sequences obviously derive from incorrect annotations (i.e. they are not real proteins). Al calculations and phylogenetic trees should be updated accordingly.

We are grateful for the comment, which helps to improve the article. Following your comments, we removed several sequences that were the most out of place with their size. Accordingly, the alignments and phylogenetic analysis were performed anew.

  1. Graphical representation of the phylogenetic trees. In general, the current representation is not clear, since the authors chose to report binary trees in all cases, regardless of the bootstrap support of the nodes, which was often very low, and collapse some portions of the tree in cartoons, even though this was unnecessary. First of all, the quality of a phylogenetic tree is based on the quality of the underlying multiple sequence alignment. As I mentioned above, it looks like the authors might have included several highly divergent, incomplete sequences in the alignment, which might have resulted in inconsistent analyses. All nodes showing poor bootstrapsupport (i.e. < 50) should be collapsed and presented as multifurcate. Representing binary nodes with support values lower than that has no biological meaning and would be indeed quite misleading. The tree rooting strategy (midpoint? User-selected? Based on an outgroup?) should be also explicitly stated. Reporting branch lengths over each branch only adds confusion: the scale bar is sufficient.

Thank you for your remark. We reanalysed all phylogenetic trees using new evolution model that was selected by IQ-TREE Web Server. Following your advice, we have collapsed most of the nodes with a low bootstrap value. Initially, the trees were built as unrooted, however, we made an error during further image processing, which we have corrected in the new version of the article. The branch length labels on your recommendation have been removed.

Round 2

Reviewer 2 Report

Unfortunately, the strategy for the identification of crystallins was merely based on available annotations, which is a severe issue for this manuscript. Functional annotations are always the product of an automated process, which can be carried out using different methods, leading to more or less accurate outcomes. Hence, the use of a unique, well-defined strategy for the identification of the members of any gene/protein family is mandatory whenever the objective of a given study is to report the evolutionary spread of such members across different taxa. Merely relying on annotations (i.e. inferred protein names) usually leads to several missed opportunities, e.g. proteins clearly belonging to the same family may not be annotated correctly, or be simply listed as "uncharacterized/hypothetical proteins" due to borderline e-values or due to their low complexity (low complexity regions are indeed usually masked during BLAST), which might have had a significant impact on the identification of crystallins in this particular case. Hence, I am sorry to say that this identification approach was not appropriate in this case, as annotation-based idenfirications should be always avoided in any evolutionary biology study or, to the very least, complemented with additional confirmatory bioinformatic identification.

Please find below some additional comments:

My comment concerning the processing of crystallins was more of a general nature: I was simply asking whether crystallins are secreted or processed in any way during their maturation, which has little to do with the way the underlying genes have been predicted. There are several bioinfomatic means by which, for example, a signal peptide can be identified.

The replacement of "medium conserved" with "semi-conserved" did not provide any improvement. This is still an arbitrary definition, which should be avoided. A protein family can be "midly conserved" or "semi-conserved" only when it is compared with a reference. If such reference is not provided, then this definition becomes abstract.

The trees still report several poorly supported nodes that should be collapsed. Nodes displaying a boostrap support lower than 50 have no biological meaning.

Author Response

Dear Reviewer,

Thank you very much for your comments, remarks and suggestions. Your contribution is very precious and important, and helps to improve the article's quality. We hope that we were able to answer all of your questions and provide you with appropriate, informative answers.  

Thank you for your time and patience,

Best regards,

Irina Dominova

Valery Zhukov.

Bellow we provide the answers for all questions.

The most significant note relates to the choice of amino acid sequences of crystallins for the phylogenetic analysis. We acknowledge the validity of this comment and the fact that the molluscan proteins we analyzed have a status "Model" in the NCBI Gene database (e.g., https://www.ncbi.nlm.nih.gov/gene/106869778; https://www.ncbi.nlm.nih.gov/gene/106076654; https://www.ncbi.nlm.nih.gov/gene/117336049). It should be noted that with respect to functional identification of crystallins, as well as most other proteins, molluscs are an understudied group of animals. This fact defines the current limit of possibilities of phylogenetic analysis of molluscan crystallins. Of course, a necessary next step should be the functional annotation and validation of these proteins, as well as the establishment of the complete gene structure, including the exon-intron structure and regulatory sequences. For validation of the amino acid sequences, of course, a large-scale proteomic analysis of all the objects in question is mandatory, which, unfortunately, is currently beyond our capabilities. In the nearest future, we plan to perform a proteomic analysis of the Pomacea lens, which should be the first step in the functional annotation of molluscan crystallins.

My comment concerning the processing of crystallins was more of a general nature: I was simply asking whether crystallins are secreted or processed in any way during their maturation, which has little to do with the way the underlying genes have been predicted. There are several bioinfomatic means by which, for example, a signal peptide can be identified.

The issue of the possibility of poststranlational processing of molluscan crystallins has not been studied at present. We cannot claim the presence or absence of these modifications due to the lack of validated crystallin sequences.

By checking the available amino acid sequences of molluscan crystallins using SignalP - 6.0 (https://services.healthtech.dtu.dk/service.php?SignalP) and DeepSig (https://deepsig.biocomp.unibo.it/welcome/default/index#:~:text=DeepSig%20is%20a%20web%2Dserver,of%20the%20query%20protein%20sequence.), we didn't identify signal peptides such as "standard" secretory signal peptides transported by the Sec translocon and cleaved by Signal Peptidase I, lipoprotein signal peptides transported by the Sec translocon and cleaved by Signal Peptidase II, Tat signal peptides transported by the Tat translocon and cleaved by Signal Peptidase I, Tat lipoprotein signal peptides transported by the Tat translocon and cleaved by Signal Peptidase II, Pilin and pilin-like signal peptides transported by the Sec translocon and cleaved by Signal Peptidase III.

The replacement of "medium conserved" with "semi-conserved" did not provide any improvement. This is still an arbitrary definition, which should be avoided. A protein family can be "midly conserved" or "semi-conserved" only when it is compared with a reference. If such reference is not provided, then this definition becomes abstract.

In view of this remark, we removed the mention of the degree of conservativity of proteins from the text. Instead, we indicated the average value of conservativity by Mirny.

The trees still report several poorly supported nodes that should be collapsed. Nodes displaying a boostrap support lower than 50 have no biological meaning.

This comment is accepted and the corresponding corrections are made to the figures 5, 7 and 9.

Round 3

Reviewer 2 Report

While I appreciate the answer provided by the authors, the fact that mollusks are non model organisms is not an excuse for the lack of an appropriate methodology for the detection of crystallin sequences.
I work with molluscan genomes every day and I am well aware of the issues linked with their annotation. For this very same reason, using available assigned gene names as the unique criteria for detecting whether crystallins (or any other gene) is present or not should never be done. There are a number of other methods, pretty simple and well within the capabilities of anybody working with sequence data, which include BLAST, the use of HMM profiles for a screening with HMMER, and many others, that would provide a much more reliable alternative. The authors are apparently not willing to improve the approach used for the identification of crystallin genes, so I regret to confirm my recommendation for rejection due to the impossibility of evaluating the reliability of the evolutionary analyses, which cover a significant part of this manuscript.

Author Response

Dear Reviewer,

Thank you very much for your comments, remarks and suggestions. Your contribution is very precious and important, and helps to improve the article's quality.

We have added a BLAST analysis of gastropod molluscs crystallins in comparison with cephalopods and bivalves. The results of this analysis are presented in Table 2.

However, it seems to us that we did not quite understand your comment regarding the annotation of mollusc genomes and the use of HMM profiles, because confirmed (validated) crystallin sequences are necessary to create such profiles. However, in this case, highly validated sequences are available only for animals phylogenetically distant from molluscs, such as zebrafish.

In this regard, it was difficult for us to fix your comment. If it is not inconvenient for you, could you please recommend us an organism or a group of organisms based on its proteins that could be used to construct HMM profiles and re-identify crystallins in molluscan genomes.

Thank you for your time and patience,

Best regards,

Irina Dominova

Valery Zhukov.

Round 4

Reviewer 2 Report

Dear authors, thank you for providing a revised version of your manuscript, which now includes some important additional data to support your finding.

I will try to explain more in detail what I meant by mentioning the use of HMMER and profiles for improving the search of crystallins in Mollusca. These analysis may go beyond the expertise or the technical capabilities of the authors, but in my opinion it would be still quite important to understand the limits of automated annotations and BLAST-based approaches, perhaps not for this study, but for others that the authors might want to carry out in the future.

I am aware about the lack of validated crystallin sequences in bivalves, which actually determines the absence of reference HMM (and correlated reference conserved domains) in dedicated repertoires such as Pfam, SMART, Interpro and others. This case is shared by several other lineage-specific protein families, which are lineage-specific and often show little homologies outside their phylum.

Yet, new HMM can be customly built by the user based on multiple sequence alignment, and these can be used to refine/improve searches for similar proteins, with much more detection power compared with BLAST, which suffers from primary sequence divergence, in particular in the case of low-complexity proteins (such as crystallins). For example, if you have a few crystallin sequences belonging to a given subfamily that you trust, these can be aligned (with MUSCLE or other MSA algorithms), and the cleaned alignment (prepared by usually trimming unaligned regions) can be used as an input for the creation of a new custom HMM profile (this is done with the hmmbuilt and hmmpress modules). Then, this newly created profile can be safely used to screen any protein database (such as NCBI nr) with hmmsearch to detect other proteins belonging to the very same family, i.e. showing a primary sequence consistent with the consensus of the crystallins included in the multiple sequence alignment. The advantages of this approach are that it does not depend on available annotations (i.e. assigned protein names) and that it is much more sensible that a regular BLAST.

I am not requiring you to carry out this demanding task in the frame of this manuscript, but to simply add some notes concerning the limits of your approach in the text. Table 2 is now useful to support the detection of significant sequence homologies between the crystallins of gastropods and other mollusks, but it also show a very high degree of variation in terms of e-value and identity level among the different hits, which makes me wonder whether the reason of these differences is technical (BLASTp usually masks low complexity regions) of has another nature (e.g. it is linked with the presence of incorrectly predicted gene models leading to incomplete proteins). Also, please note that the detection of a significant homology between A and B, does not necessarily mean that A and B are orthologous, and neither that B belongs to the very same protein family of A. For example, it is unclear to me whether the different crystallin subfamilies share any sequence similarity with each other or not. Moreover, some e-values are somewhat borderline (I see, for example a “1e-2”), which is not particularly reassuring when it comes to a reliable identification of sequences belonging to the same protein families.

The bottom line is that, considering the lack of previous characterization of molluscan crystallins, their proper identification from different molluscan classes is a challenging task and that annotation or even sequence homology based detection methods suffer from important limitations. In light of these considerations, the sequences reported in this manuscript may be just a fraction of the full repertoire of crystallins present in these organism, and others, perhaps more divergent, may remain to be discovered. For the very same reason, the annotation of several crystalline genes is unreliable, leading to truncated proteins (the authors have clearly seen this through the multiple sequence alignments they performed), which further complicates phylogenetic inference.

Please update the text accordingly. With table 2 in place, I think that improving the critical discussion about the limitations of this study in terms of elucidating the evolutionary history of these genes will be sufficient to make this work suitable for publication at this stage.

Author Response

Dear Reviewer,

Thank you very much for your comments, remarks and suggestions. Your contribution is very precious and important, and helps to improve the article's quality.

We are very grateful for your clarification on ways to re-annotate molluscan genomes, and we will certainly take your advice for further studies. Furthermore, we already have an idea how your recommendation can be used to search for some proteins in the Achatina genome. Thanks again for this explanation.

Also, at your recommendation, we updated the "Discussion" section with relevant descriptions concerning the results obtained during the BLAST analysis (Table 2). Moreover, at the end of the article, we highlighted that all our results are preliminary, as they are based only on bioinformatic approaches, and require verification by, for example, transcriptomic or proteomic analysis.

Thank you for your time and patience,

Best regards,

Irina Dominova

Valery Zhukov.
